# Potential for Economic Transition and Key Directions of Cross-Border Cooperation between Primorsky Krai (Russia) and Jilin (China)

**Mengchen Ji [1,2], Fujia Li [1], Shuangjie Xu [1,3], Yan Zhuang [1,2,*], Tsydypov Bair [4], Alexey Bilgaev [1,4] and Kexin Guo [1,2]**

[1] Institute of Geographic Sciences and Natural Resources Research, Chinese Academy of Sciences, Beijing 100101, China
[2] University of Chinese Academy of Sciences, Beijing 100049, China
[3] School of Geographical Sciences, Liaoning Normal University, Dalian 116029, China
[4] Baikal Institute of Nature Management, Siberian Branch, Russian Academy of Sciences, 670047 Ulan-Ude, Russia
[*] Correspondence: zhuangyan@cashq.ac.cn

**Abstract:** The potential of cross-border regional cooperation development is important for evaluating the present and future capacity for regional development. Primorsky Krai (Russia) and Jilin (China) share borders and complementary advantages. China and Russia have growing intention to strengthen cooperation, with geopolitical considerations. This study constructs a regional development potential assessment index system from the perspective of cross-border cooperation, and it establishes a multilayer cross-border regional development potential assessment (RDPA) model with three layers: target, factor, and indicator. By coupling four modules (economic, social, transportation, and resource environment), this study builds a development potential value system. The results illustrate that Primorsky Krai and Jilin have potential for cross-border cooperative development. The results reveal that Primorsky Krai has comparative advantages in port logistics, agriculture, fishery, and mining, which are all complementary to Jilin. These are important directions for cross-border international cooperation. Additionally, this study provides suggestions on decision-making and strategy consultation for the international cooperation between Jilin and Primorsky Krai and creates references for the study of cross-border cooperation opportunities and challenges for other neighboring regions around the world.

**Keywords:** economic transition; cross-border cooperation; Jilin; China; Primorsky Krai; Russia

## 1. Introduction

Regional development potential is an important standard for evaluating the capacity for regional development. Recently, international cooperation has become increasingly crucial in the context of globalization; thus, it is more necessary to evaluate regional development potential from the perspective of cross-border cooperation. Primorsky Krai is a node of the China–Mongolia–Russia Economic Corridor and has an important geopolitical and economic position in the implementation of China's "Belt and Road" initiative [1]. Jilin, in the northeast of China, is an important strategic area for China's development, with many long-established industrial bases. With geographical proximity and complementary advantages, the two regions have broad prospects for cross-border cooperation. Cross-border cooperation potential research is based on the study of regional development potential, featuring typical cross-border regions, with Primorsky Krai–Jilin as a key region of cross-border regional cooperation. The present study focuses on three categories: cross-border regional cooperation, studies of regional development transformation potential, and regional development strategies in the Russian Far East. Generally, cross-border cooperation is a type of territorial cooperation of different units in border areas. Common

historical origins and other social and/or economic similarities (for example, linguistic, cultural, or constitutional) of neighboring regions may help to achieve the common development goals [2]. Cross-border cooperation is recognized as being helpful to develop border regions' economy [3,4], with the principle of equal resources, responsibilities, risks, and benefits [5]. Brunet-Jailly (2022) provided an overview of cross-border coordination and cooperation across global continents [6], arguing that regional drivers determine the type of this relationship. Kurowska-Pysz (2017) provided a comparative analysis of the Polish–Lithuanian border and assessed the sustainability factors in Czech–Polish cross-border cooperation [7]. Jose (2022) analyzed cross-border governance in the Amazonian cross-border region among Peru, Brazil, and Bolivia from a systematic perspective and explored cross-border cooperation models in the border regions of southern Finland and Estonia [8]. In general, cross-border cooperation research mostly focuses Europe. The differences between cross-border regions and the motivations for cooperation influence cross-border cooperation [9–11]. The large differences between China and Russia in economic development, the obvious complementary characteristics, and the great willingness to cooperate in the cross-border economy impart a high value of cross-border economic cooperation research between Primorsky Krai and Jilin Province. Regional development potential studies focusing on the potential for economic development transformation [12] are divided into qualitative studies [13–15], indicator composition studies [16,17], and evaluation methods studies [18,19]. Regional economic transformation is a process of institutional reform and institutional innovation that provides a new incentive structure for economic growth and the internalization of external effects [20–22]. Cross-border regional cooperation includes the idea of geographies of transition, focusing on the multiscale geographic co-evolution of emerging transitional development and the dynamics of states' relations and the unbalanced regional development behind them, and it has become an important part of "evolutionary economic geography" [23–27]. However, transformation geography research has paid less attention to developing countries—especially the local transformation practices in China within the global transformation system [28]. The existing studies primarily focus on single case analysis [29–31] and ignore regional comparative analysis. Regional development strategy research in the Far East includes regional development, demographic issues, economic development, resource development, etc. [32]. Insufficient studies on the economic transformation of separate first-level administrative regions have been conducted. Therefore, it is highly anticipated that a better study of cross-border economic–geographical transformation cooperation between the neighboring regions of Russia and China should be conducted, and Primorsky Krai and Jilin Province are key areas for both countries considering their own interests. Research directions of development cooperation between China and Russia include development policy research [33–36] and regional economic cooperation research [37–40]. China–Russia regional cooperation has a strong basis. The two countries have common interests to pursue, and the northeast region of China—as one of the key regions for cooperation with Russia—can promote regional cooperation by establishing a China–Russia community of shared interests.

It is true that relevant studies have made some progress, but there is still huge potential to achieve. On the one hand, the research on the Sino–Russian Far East and Northeast cross-border regions is insufficient. Russia is currently trapped in the Russian–Ukrainian conflict, facing serious geopolitical and economic problems, and blocked by the west. At the same time, there is typical unbalanced regional development within Russia, where the Far East has been lagging behind for a long time. Primorsky Krai is a better choice for Russia to open up the development pattern of the Far East both in terms of location conditions and on a trade basis. Moreover, China is facing international trade friction with the United States, while its northeastern region, represented by Jilin, is facing severe, continuous low economic growth and a relatively lagging level of opening up to the outside world. Therefore, it is important to carry out cross-border cooperation research on Primorsky Krai–Jilin as a key area in the region. This is an inevitable choice in the interests of both countries. On the other hand, the existing studies mostly focus on one country or one region, lacking

a comprehensive comparative analysis of the two neighboring regions. The study of the development potential of Primorsky Krai–Jilin cross-border regional cooperation can be a good complement to the study of cross-border cooperation.

This study constructs an evaluation model of the development potential of cross-border regions in Russia and China. The model is set at the provincial level and is used to make a comprehensive comparison of the development potential of the two regions in economic, social, transportation, and resource environment terms. This paper analyzes the regional complementarity, explores the prospects of cross-border cooperation, and proposes a system of cross-border development potential indicators and data. This paper also analyzes the first-hand data obtained from the research team's fieldwork and integrates multiple subsystems of Russia and China. In addition, the evaluation model provides a comparative and integrated means of analysis for different regions in different countries, and a comprehensive comparative analysis of the results is conducted, with improvement on the basis of individual regions. This study can provide support for decision-making and strategic advice for international cooperation practices in Jilin and Primorsky Krai. Additionally, it can be instrumental to the future study of cross-border cooperation potential and countermeasures in other neighboring regions around the world.

In general, this study compares key regions of cross-border cooperation. The study enriches research methods for evaluating the development potential of cross-border regions and provides scientific and technological support for promoting the economic development of the Russian Far East and the long-established industrial bases of Northeast China.

## 2. Materials and Methods

### 2.1. Research Area

Primorsky Krai is one of the 85 regional subunits of Russia. It is in the southernmost part of the Far East (Figure 1), next to Heilongjiang and Jilin of China to the west. Primorsky Krai, with a coastline of about 1500 km, is an important sea transportation hub. The "Binhai 1" and "Binhai 2" International Transport Corridors (ITCs) connect the Chinese seaports of Heilongjiang and Jilin to Primorsky Krai, making it easier for people to work together across borders. Primorsky Krai's economy is based on fishing, forestry, mining, and ship repair. It is also a source of raw materials for non-ferrous metalworking, wood processing, and fish processing. Primorsky Krai has a lot of mineral resources, including over 3.7 billion tons of coal. Therefore, Primorsky Krai's good location and natural resources give it huge potential for growth and cross-border cooperation.

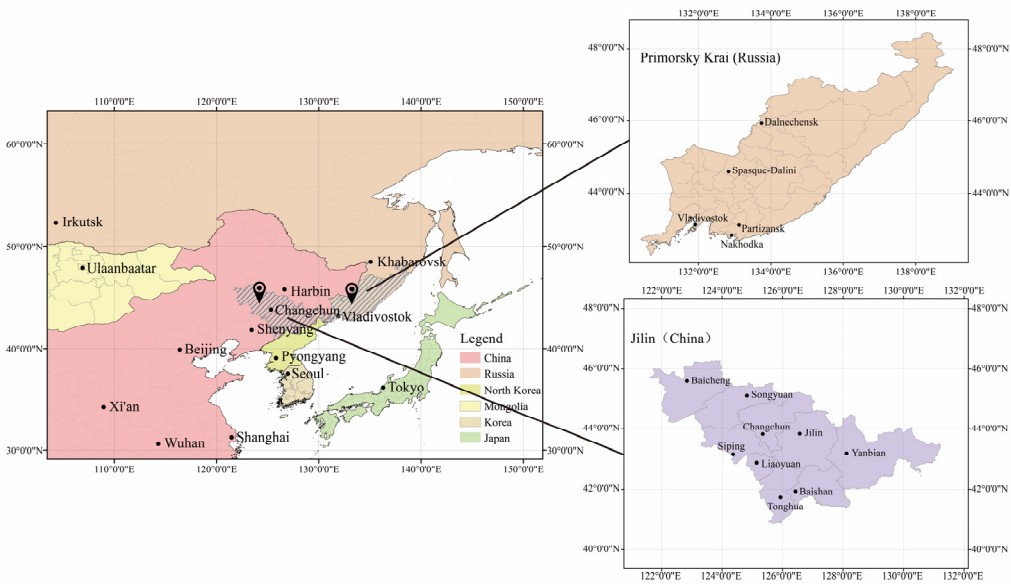

**Figure 1.** Location of Primorsky Krai and Jilin.

Jilin is a province of the People's Republic of China (Figure 1). It is located near the sea and shares borders with Primorsky Krai (Russia) and North Korea. This makes it a good place for international trade and exchange. Jilin's long-established industrial bases have an opportunity to reboom, especially with its well-developed processing and manufacturing industry. Jilin is also an important place for producing grains for the whole country. It is in the world-famous "Golden Corn Belt" and "Golden Rice Belt". Jilin is the leading corn exporter in the whole country. In terms of international trade and economic cooperation, Jilin has officially established ties with 155 countries and regions around the world, and 89 of the world's top 500 multinational companies have ventures in Jilin. Since the developing and opening-up strategy of the Changchun–Jilin City–Tumen pilot area was put into place, the international road transportation network has started to take shape. This network connects Russia and North Korea and spreads out to Northeast Asia. Jilin is in the geographic center of Northeast Asia and has the best conditions for cooperation with other Northeast Asian regions. Therefore, Jilin potentially dominates the cooperation between China and Russia in the region and, of course, is the best counterpart of Primorsky Krai.

China's "Belt and Road Initiative" and Russia's "Look East" fit one another. This is an important policy basis for Jilin and Primorsky Krai to achieve their cross-border potential. At present, trade between the two regions keeps soaring, and closer economic and trade cooperation is underway (Figure 2). The East–West economic and trade conflict is still ongoing because of geopolitical factors, but China and Russia have the potential to work more closely. Moreover, China's Northeast Revival Plan matches with Russia's Far East development strategy to a great extent. Consistent regional cooperation plans are serving as guidelines for cross-border cooperation to achieve the growth potential.

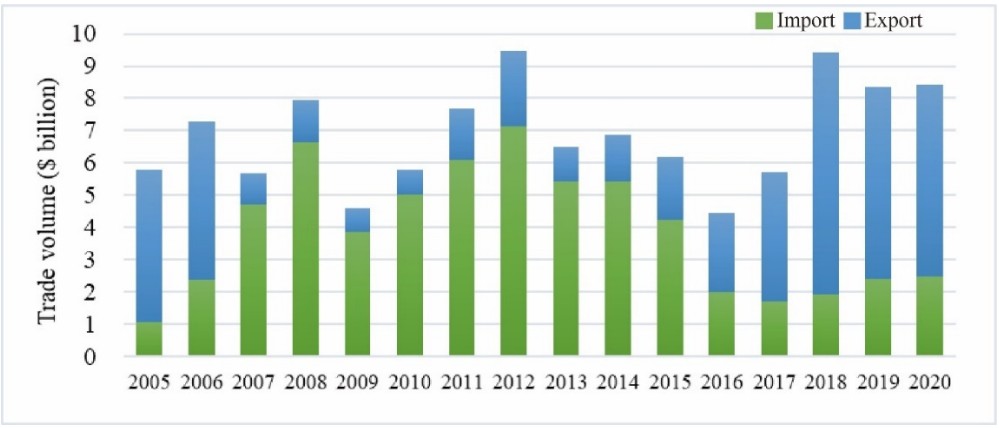

**Figure 2.** Trade between Jilin Province and Russia.

*2.2. Methods*

2.2.1. Indicator System

This research extensively exploits and analyzes the data on development potential. Our evaluation index system is built on cross-border regional complementarity and China–Russia cooperation. This research builds a development potential evaluation index system with three layers: a target layer, a guideline layer, and an indication layer, as shown in Table 1.

The index system consists of 4 targets, 21 guidelines, and 45 indicators, providing a comprehensive and specific quantitative evaluation of the regional development potential. Due to the differences in statistical methods and statistical calibers between China and Russia, this study works on the correspondence of the two similar indicators as much as possible. The specific correspondence is listed in Table 1, with the relevant indicators of Primorsky Krai on the left and the relevant indicators of Jilin Province corresponding to each of the former indicators on the right. The target-level indicators consist of four items (economy, society, transportation, and resources and environment), which can summarize

the overall socioeconomic situation and are equally important components for measuring the potential of economic transformation. The economic development level reflects the comprehensive economic development of the region from the perspective of development potential through five aspects, represented by total economic volume, economic quality, investment and trade, income and consumption, and finance. The social development level reflects social services and social development through five parts, including population, urbanization, social burden, education, and medical care. The transportation level reflects the regional transportation strength and convenience, including road network density, transportation volume, and turnover. The resource environment reflects the ecological advantages of resources in the region; for the convenience of computing, this study uses the values of statistics such as total forestry output value, total agricultural output value, etc., and the vacant data are similarly replaced to ensure the consistency of these domestic data. All relevant data are from 2019, avoiding the socioeconomic disturbance of the data in 2020 due to COVID-19, and the chronology is similar, accurate, and representative. The overall index system for evaluating the development potential of Primorsky Krai and Jilin provides a comprehensive and specific quantitative evaluation of the regional development potential.

**Table 1.** Evaluation index system of the development potential of Primorsky Krai and Jilin.

| | | Primorsky Krai | Jilin | |
|---|---|---|---|---|
| **Target Layer** | **Guideline Layer** | **Indicator Layer** | | **Indicator Direction** |
| Economic | Total economy | Gross domestic product | Gross domestic product | + |
| | | Per capita regional GDP | Per capita regional GDP | + |
| | Economic quality | Gross agricultural product | The added value of primary industry | + |
| | | The output value of the manufacturing industry | The added value of secondary industry | + |
| | | Turnover of public catering | The added value of the tertiary industry | + |
| | | Fixed capital investment | The growth rate of fixed assets investment | + |
| | Investment and trade | Foreign capital investment | Total imports of foreign-funded enterprises | + |
| | | | Total export volume of foreign-funded enterprises | + |
| | | Import | Import | + |
| | | Exports | Export | + |
| | | Retail trade volume | Retail business income | + |
| | | Wholesale trade volume | Operating income of the wholesale industry | + |
| | Income and consumption | Per capita income | Per capita income | + |
| | | Per capita consumption expenditure | Per capita consumption expenditure | + |
| | | Price consumption index | Price consumption index | − |
| | Finance | Comprehensive budget income | General budget revenue | + |
| | | Comprehensive budget expenditure | General budget expenditure | + |
| Social | Area | The measure of area | Urban area | + |
| | Population | Total population | The permanent population at the end of the year | + |
| | | Natural population growth rate | Natural population growth rate | + |
| | | Migration growth per 10,000 population | Inflow population | + |
| | Urbanization | The proportion of the urban population | The proportion of the urban population | + |

**Table 1.** *Cont.*

| | | Primorsky Krai | Jilin | |
|---|---|---|---|---|
| Target Layer | Guideline Layer | Indicator Layer | | Indicator Direction |
| Social | Social burden | Non-working-age population per 1000 working-age population | Total dependency ratio | − |
| | | Unemployment rate | The registered urban unemployment rate | − |
| | Educational research | Number of undergraduate and junior college students | The average number of college students per 100,000 population | + |
| | | Number of R&D personnel | Number of R&D personnel | + |
| | | Number of R&D organizations | Number of R&D organizations | + |
| | | R&D capital expenditure | R&D capital expenditure | + |
| | Medical coverage | Number of beds per 10,000 people | Number of health technicians per 10,000 people | + |
| | | Number of doctors per 10,000 people | Number of beds in medical institutions per 10,000 people | + |
| Transportation | Transportation infrastructure | Railway track density | Railway operating mileage | + |
| | | Highway density | Highway mileage | + |
| | | Number of ports | Number of ports | + |
| | | Number of airports | Number of airports | + |
| | Shipping volume | Passenger traffic volume | Passenger traffic volume | + |
| | | The volume of freight transport | The volume of freight transport | + |
| | Turnover | Passenger turnover | Passenger turnover | + |
| | | Cargo turnover | Cargo turnover | + |
| Resources and environment | Forestry | Forest coverage | The total output value of forestry | + |
| | Agricultural | Crop sown area | The total output value of animal husbandry | + |
| | Livestock | Production weight of livestock and poultry for slaughter | The total agricultural output value | + |
| | Fishery | Catch | The total output value of the fishery | + |
| | Mineral | Mineral output value | Mineral output value | + |
| | Energy | Energy output value | Energy output value | + |
| | Environment | Wastewater discharge | The total amount of urban sewage discharge | − |
| | | Emission of air pollutants | Sulfur dioxide emissions | − |

### 2.2.2. Statistical Methods

Quantitative assessment of cross-border development potential is a complex process that needs to consider the economy, society, transportation, and resource environment as a whole, and each of these four elements has different data to give it a value and judgement. Therefore, to conduct a cross-border assessment of a region's potential, we calculate the information entropy as the basis for a comprehensive evaluation of the discrete degree of indicators [41–45]. Furthermore, we use the aggregation function to assign aggregated values to the contribution of relevant decision variables to determine the overall value of the region [46,47]. This study takes into account the qualitative factors that cannot be predicted, and it also uses the network of the International Union of Scientists along the Belt and Road region, which was set up by our research team, to find the model through expert consultation. More than 10 experts were asked to rate the target layer weights' growth potential online in the same set of comments. They also responded with their thoughts on the cooperation of Primorsky Krai and Jilin. The Delphi method was used in this study to figure out the weights based on the experts' opinions. We also refer to the research models

and methods of Li [39], Liu [48], and Yager [49,50], and all of these studies help in building the regional development potential evaluation model.

The method is as follows:

(1) Economic, social, transportation, and resource and environmental potential evaluation module.

We start by making a data matrix with $n$ economic, social, transportation, and resource and environment indicators for $m$ provincial administrative regions ($X$) $\begin{bmatrix} x_{11} & \cdots & x_{1n} \\ \vdots & \ddots & \vdots \\ x_{m1} & \cdots & x_{mn} \end{bmatrix}$,

where $x_{ij}$ is the value of the sample ($i$) of the $j$ evaluation ($j$) indicator in matrix $X$, $0 \le i \le m$, and $0 \le j \le n$.

We standardize the evaluation metrics to eliminate the effect of the magnitude.

$$\begin{cases} x_{ij} = \frac{x_j - x_{min}}{x_{max} - x_{min}} \text{ Positive indicators} \\ x_{ij} = \frac{x_{max} - x_j}{x_{max} - x_{min}} \text{ Reverse Indicator} \end{cases} (0 \le i \le m, \ 0 \le j \le n). \tag{1}$$

where $0 \le i \le m$, $0 \le j \le n$, $x_{max}$ is the maximum value of the item $j$ indicator, $x_{min}$ is the minimum value of the item $j$ indicator, and $x_{ij}$ is the standardized value.

Then, the entropy value of the $j$th indicator can expressed as follows:

$$e^j = -k \sum_{i=1}^{m} S_{ij} ln S_{ij}. \tag{2}$$

where $k$ is a constant, and $S_{ij}$ is the weight of the sample ($i$) value under the indicator ($j$) to that indicator.

Then, we obtain the sample's evaluation scores $c_1$, $c_2$, $c_3$, and $c_4$ for the economic, social, transportation, and resource environments.

$$\begin{cases} c_1 = \Sigma_{j=1}^{n} w_j x'_{ij} \text{ Economic indicators layer} \\ c_2 = \Sigma_{j=1}^{n} w_j x'_{ij} \text{ Social indicator layer} \\ c_3 = \Sigma_{j=1}^{n} w_j x'_{ij} \text{ Traffic state indicator layer} \\ c_4 = \Sigma_{j=1}^{n} w_j x'_{ij} \text{ Resource and environmental indicators layer} \end{cases} \tag{3}$$

where $w_j$ is the weight of the indicator ($j$).

(2) Calculation of target layer weights.

In this study, experts were asked to evaluate the significance of the target layer using a 5-point Likert scale (5—very important; 4—relatively important; 3—somewhat important; 2—generally important; 1—unimportant). The scale was calculated as follows:

$$\overline{f_q} = \frac{1}{a} \sum_{i=1}^{a} F_{qi}. \tag{4}$$

where $a$ is the number of experts, $F_{qi}$ is the score of expert ($i$) for target layer ($q$), and $\overline{f_q}$ is the average of all experts' scores for target layer ($q$). The weights ($w_q$) of the target layer ($q$) can be calculated as follows:

$$w_q = \overline{f_q} \frac{1}{\sum_{q=1}^{4} \overline{f_q}} \tag{5}$$

(3) Comprehensive development potential evaluation.

The four subsystems are put together to conduct a full evaluation of regional development based on the evaluations of economic, social, transportation, and resource and environmental potentials. The method is as follows:

We normalize the initial score of each target layer:

$$c'_q = \frac{c_q - c_{min}}{c_{max} - c_{min}}.\tag{6}$$

where $c_q$ is the target layer ($q$) initial score, $c'_q$ is the standardized score of target layer ($q$), and $q$ is the target-level code for development potential evaluation, where $1 \leq q \leq 4$. $c_{max}$ is the maximum value of $c_q$, and $c_{min}$ is the minimum value of $c_q$. The comprehensive level of development potential ($H$) of each district can be expressed as follows:

$$H = \sum_{q=1}^{4} c'_q w'_q.\tag{7}$$

(4)　Analysis of the Development Potential of Primorsky Krai–Jilin Cross-border Regional Cooperation.

Through steps 1 and 2, the scores of cross-border cooperation and regional development potential for Primorsky Krai and Jilin can be figured out. For cross-sectional analysis and research, we compared the two study areas. We used the comparative relativity method to pick 21 criterion-level indicator scores and figured out the differences between the same indicators in the two regions. The method was as follows: first, the criterion-level indicators for the coastal border area were labeled $c_{ir}$, and the criterion-level indicators for Jilin were labeled $c_{ic}$. Next, the data for the same indicators in different regions were turned into percentages. This gave us the comparative relative number ($N_i$) of the indicators of the development potential of Primorsky Krai–Jilin cross-border regional cooperation, which can be expressed as follows:

$$N_i = (c_{ir} \div c_{ic}) \times 100\%.\tag{8}$$

When $N_i > 100\%$, Primorsky Krai has a stronger potential for cross-border regional development cooperation in indicator ($i$). When $N_i = 100\%$, Primorsky Krai and Jilin have equal potential for cross-border regional development cooperation in indicator ($i$). When $N_i < 100\%$, Jilin has a stronger potential for cross-border regional development cooperation in indicator ($i$).

### 2.2.3. Data Sources

This study is part of the national science and technology basic resources survey special project "China-Mongolia-Russia International Economic Corridor Multidisciplinary Joint Study". In this project, we have made trips to Russia to conduct field research and obtained a large amount of first-hand research data, which are unique in terms of data information. Furthermore, the data on Russia's development potential include the *Russian Statistical Yearbook 2020*, the Russian Far East Borderlands Development Program 2015–2025, and the Strategy for Socio-Economic Development of the Russian Far East and Baikal Region up to 2025. The data on China's development potential were obtained from the website of the National Bureau of Statistics of China, *the China Statistical Yearbook on the Environment*, the *China Social Statistics Yearbook*, the *China Energy Statistics Yearbook*, the *China Population Census Yearbook*, etc. To ensure the similarity and consistency in the selection of development potential evaluation indicators between Primorsky Krai and Jilin, unit processing and similar substitution were carried out for individual indicators to meet the research requirements.

### 3. Development Potential Analysis
#### *3.1. Economic Development Potential Analysis*

According to the regional development potential assessment (RDPA) model, we separately calculated the economic development potential scores for each first-level administrative region in Russia and China, and we subsequently compared and analyzed the results, which are shown in Figures 3 and 4.

The researchers utilized the economic development potential scores to classify the 85 first-level administrative regions of Russia and 31 first-level administrative regions of China (excluding Taiwan Province, the Hong Kong Special Administrative Region, and the Macao Special Administrative Region) into five levels based on their degree of economic potential, namely, I (highest), II (high), III (medium), IV (low), and V (lowest). As per the results, Primorsky Krai ranked 29th among all 85 first-level administrative regions. It was categorized as having high economic development potential, with a score of 0.803. In contrast, Jilin ranked 25th among all administrative regions and was classified as having low economic development potential, with a score of 2.589.

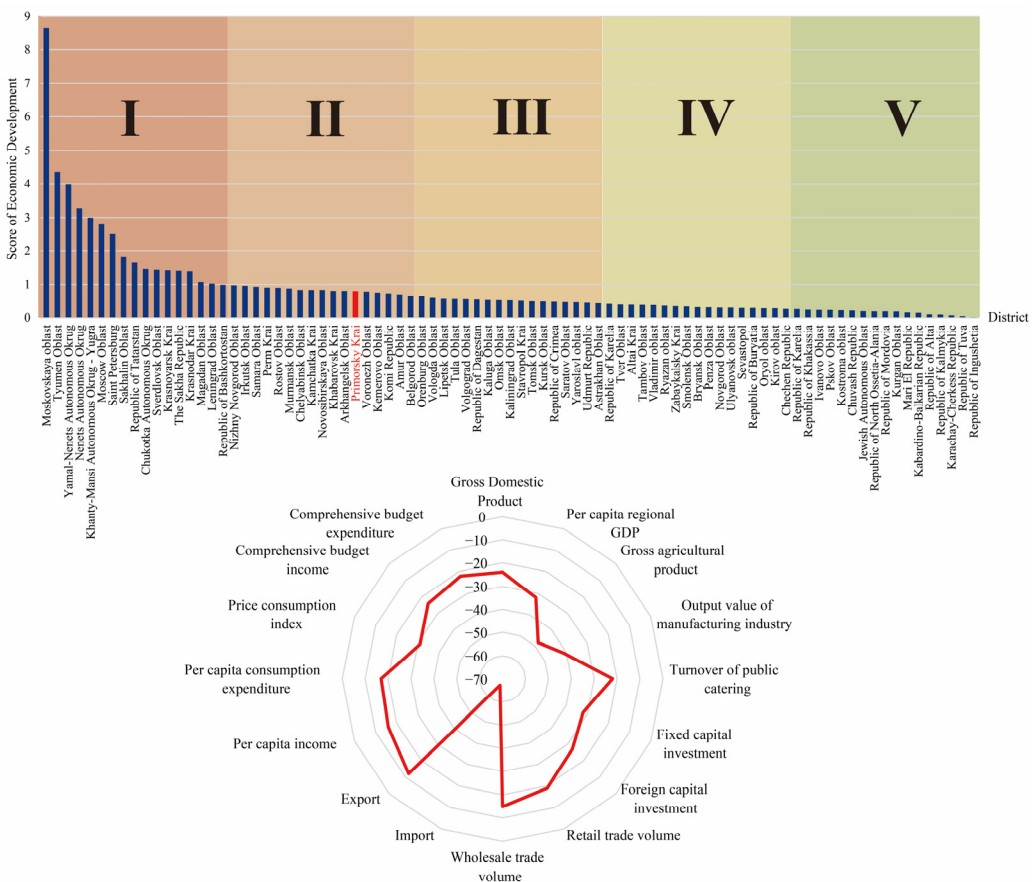

**Figure 3.** Economic development potential cores of Primorsky Krai and ranking by indicators. (I—highest, II—high, III—medium, IV—low, and V—lowest).

This study selected 16 tertiary indicators to measure the economic development potential of Primorsky Krai, while 17 tertiary indicators were chosen for Jilin. This study assessed the economic development potential from five perspectives: economic volume, economic quality, investment and trade, income and consumption, and finance. To make them easier to read in the radar chart, the ranking of each indicator was assigned a negative value. This means that the farther away from the center of the circle, the higher the ranking and the higher the development potential. The top three indicators of economic development potential in Primorsky Krai are exports, wholesale trade, and per capita consumption expenditure, ranking 12th, 15th, and 17th respectively. These are important factors in enhancing the economic development potential of the region. The consumer price index and total imports of foreign-invested enterprises are the top two indicators of Jilin's economic development potential; they are ranked 8th and 15th, respectively, and are important factors to enhance the economic development potential of the region. The index of total imports of foreign-invested enterprises in Jilin is excellent, which corresponds well with the export index of Primorsky Krai. The two sides have a good basis for carrying out

import and export trade cooperation, which can result in mutual benefits and win–win situations. However, Primorsky Krai ranks last in terms of gross agricultural product (48th) and imports (67th). Primorsky Krai is mostly made up of mountainous areas (80%) and some plains, making it hard for agriculture to grow. In addition, Primorsky Krai mainly focuses on import trade. The export trade has large room for improvement with the development of the regional economy and the construction of ports in Primorsky Krai. Jilin ranks 22nd in terms of the added value of primary industry, and Primorsky Krai ranks lower. The two are complementary in terms of primary industry development and have good prospects for cross-border cooperation.

> "Comparing the indicators of economic development between Primorsky Krai and Jilin, we can see that all criteria-level indicators of Jilin's economic development potential are better than those of Primorsky Krai, with the advantages being more obvious in the fields of economic quality and investment and trade. The gap between the two sides is smaller in income and consumption. Jilin can leverage its economic advantages to actively promote cross-border economic cooperation with Primorsky Krai to achieve a win-win situation".

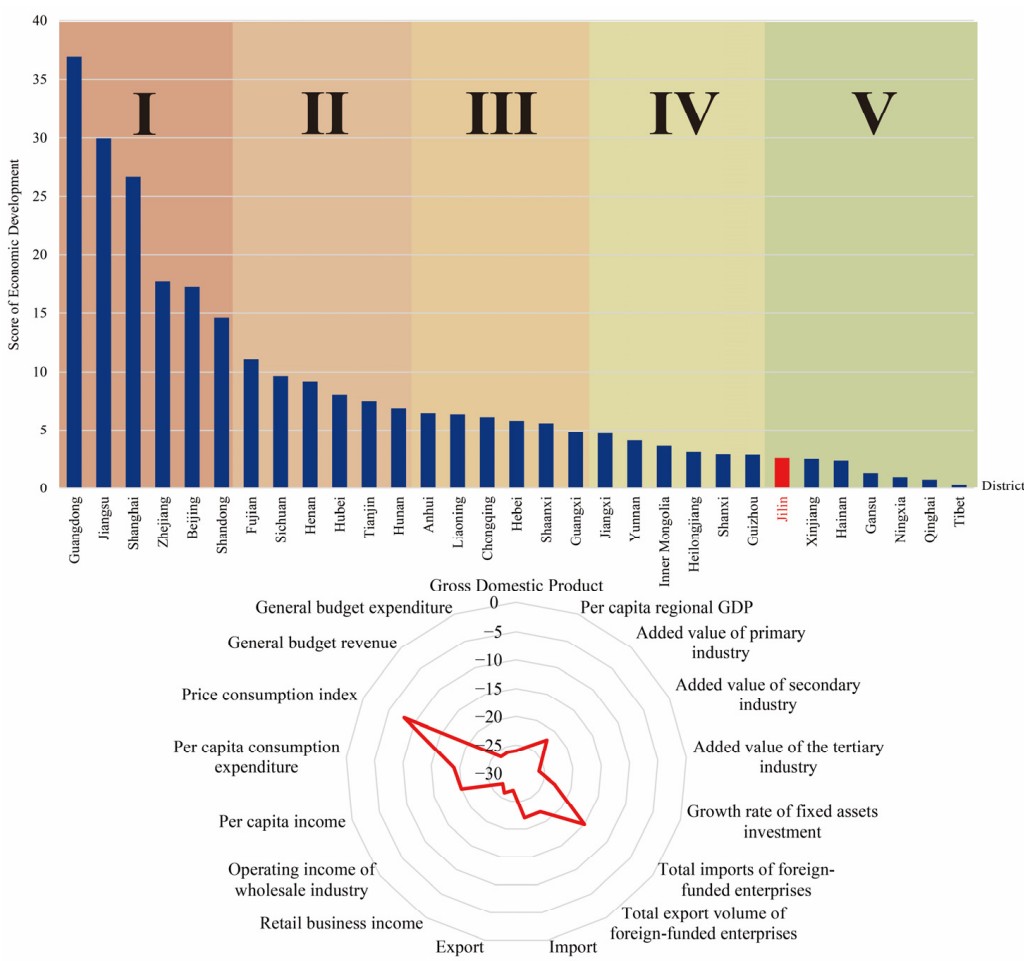

**Figure 4.** Jilin economic development potential score and ranking by indicators. (I—highest, II—high, III—medium, IV—low, and V—lowest).

### 3.2. Social Development Potential

The social development potential scores of each first-level administrative region in Russia and China were separately calculated using the regional development potential assessment (RDPA) model, and the results are displayed in Figures 5 and 6. Based on the social development potential scores, we classified 85 first-level administrative regions in

Russia and 31 first-level administrative regions in China (excluding Taiwan Province, the Hong Kong Special Administrative Region, and the Macao Special Administrative Region) into five levels of I, II, III, IV, and V, indicating the highest, high, medium, low, and lowest economic development potential, respectively. The results showed that Primorsky Krai ranks 31st among all 85 first-class administrative regions. This indicates that Primorsky Krai belongs to the administrative regions with high social development potential. Jilin ranks 26th among all 31 first-class administrative regions. Jilin is therefore classified as a region with high social development potential.

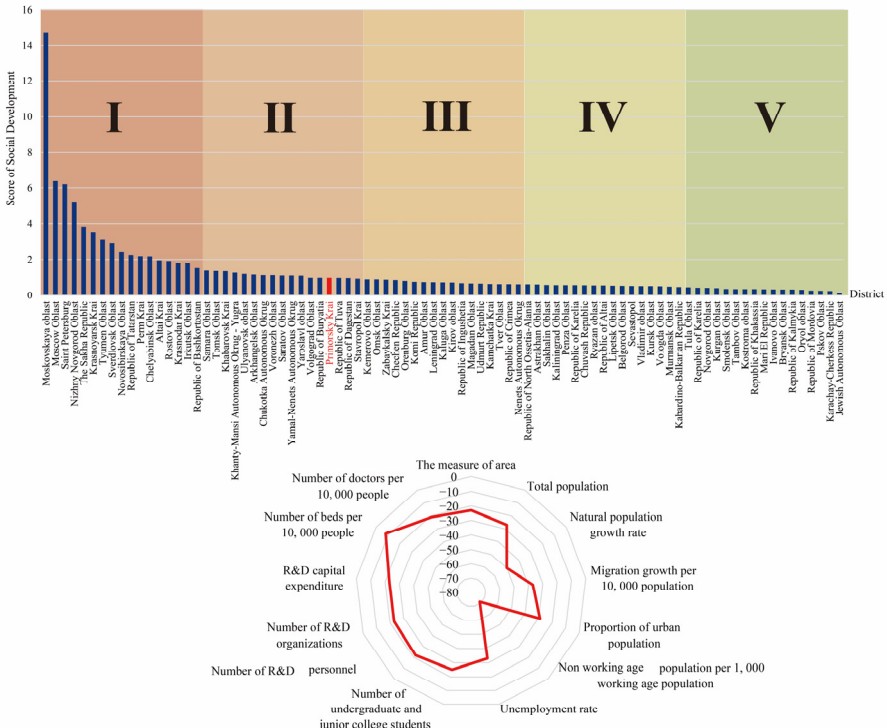

**Figure 5.** Social development potential score of Primorsky Krai and ranking of each indicator. (I—highest, II—high, III—medium, IV—low, and V—lowest).

The social development potential of Primorsky Krai and Jilin was determined from 13 corresponding tertiary indicators. These indicators measured six aspects: area, population, urbanization level, social burden, education and research, and medical security. To make the radar chart easier to understand, each indicator's ranking was given a negative value. This means that the higher the ranking and the development potential, the farther away from the center of the circle the indicator will be. The number of hospital beds per 10,000 people is the most important indicator of social development potential in Primorsky Krai, and it ranks 8th in the country. According to the data of the *International Statistical Yearbook 2019*, the average number of hospital beds per 10,000 people in high-income countries is 41. Primorsky Krai has reached 98 beds, which exceeds the standard for high-income countries. Therefore, Primorsky Krai can afford to develop faster from a medical point of view. Other indicators are more concentrated in the range of 20th to 25th place, e.g., the number of doctors per 10,000 people (21st), the number of R&D personnel (22nd), the number of R&D organizations (23rd), R&D capital expenditures (23rd), area (23rd), and the number of undergraduate students (25th). Primorsky Krai has significant strengths in healthcare coverage, education, and research. These strengths provide a strong guarantee for higher development potential. The highest-ranking indicator of Jilin's social development potential is the "average number of students in higher education", in which it ranks 5th in the country. However, R&D-related indicators are relatively low, and the capacity for science and technology innovation is insufficient. Meanwhile, Primorsky Krai

has a significant advantage in scientific research, which is complementary to Jilin. The shortage of labor in Primorsky Krai is an issue that can be resolved through international cooperation and other means, leading to an improvement in the region's development potential. Jilin's larger population could make up for the shortage of workers in Primorsky Krai by sending workers across the border.

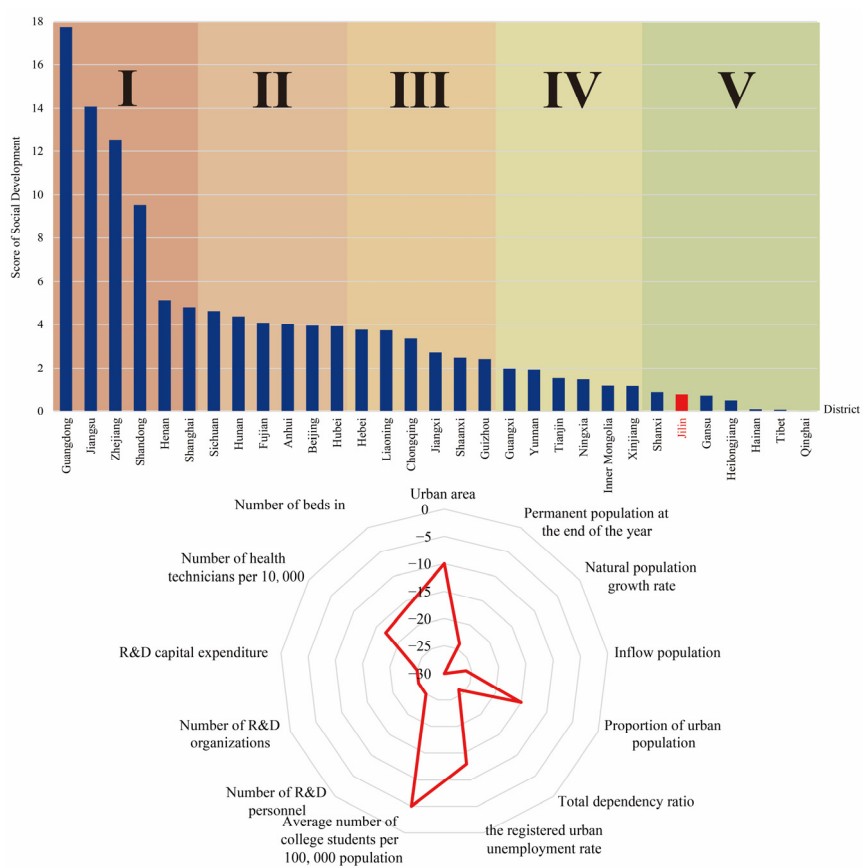

**Figure 6.** Jilin social development potential score and ranking by indicators. (I—highest, II—high, III—medium, IV—low, and V—lowest).

A comparison of the social development indicators at the criterion level between Primorsky Krai and Jilin showed that Jilin has a better social development potential overall. These indicators include area, population, urbanization, social burden, education and research, and medical security. While Primorsky Krai has a lower social burden than Jilin, it is slightly weaker in terms of education and research. This creates an opportunity for academic exchanges and research cooperation between the two regions, especially in the field of education and research. The difference between the two regions allows for potential investment in regional cross-border cooperation.

### 3.3. Transportation Potential Analysis

The RDPA model was used to calculate the transportation development potential scores of each first-level administrative region in Russia and China separately. The results of the calculations are presented in Figure 7 for Russia and in Figure 8 for China. Based on the traffic development potential scores, the researchers classified 85 first-level administrative regions of Russia and 31 first-level administrative regions of China (except for Taiwan Province, the Hong Kong Special Administrative Region, and the Macao Special Administrative Region) into five levels of higher, high, medium, low, and lower. These levels are shown by different colors and labeled I, II, III, IV, and V, respectively. The results show Primorsky Krai and Jilin's transport development potential according to the quantile classification method in Figures 7 and 8, respectively.

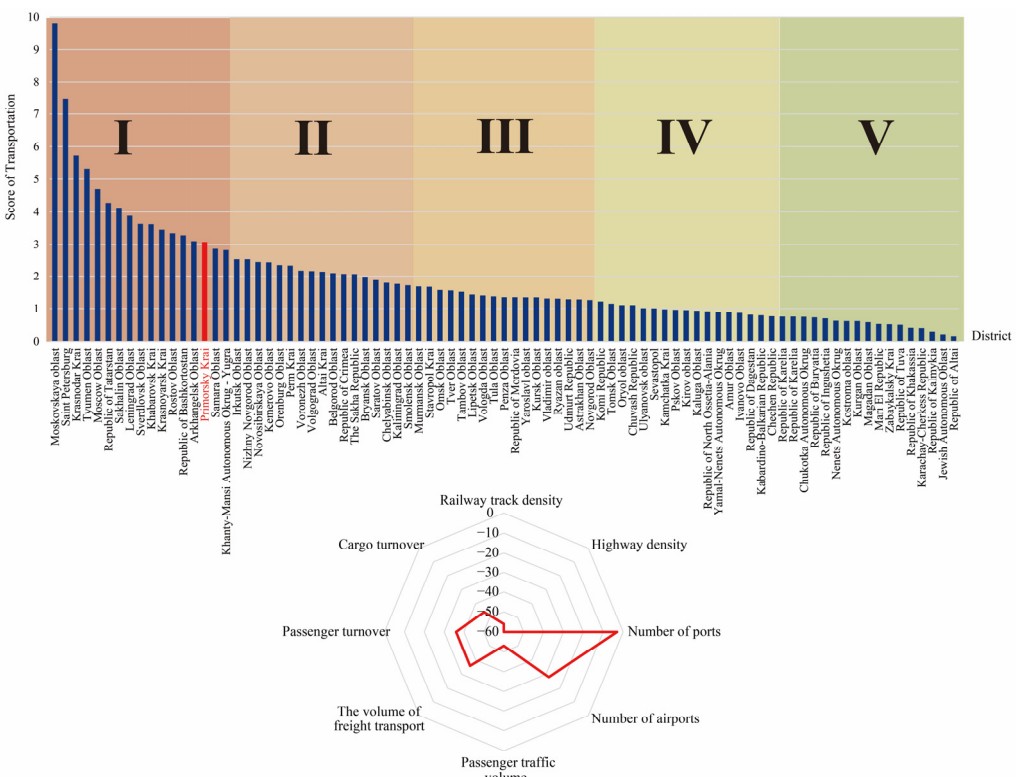

**Figure 7.** Transport development potential score of Primorsky Krai and ranking of each indicator. (I—highest, II—high, III—medium, IV—low, and V—lowest).

The results show that Primorsky Krai ranks 15th among all 85 first-class administrative regions. This indicates that Primorsky Krai belongs to the administrative regions with high transportation development potential. On the other hand, Jilin ranks 25th among all administrative regions. This indicates that Jilin belongs to the administrative regions with low transportation development potential at present.

Primorsky Krai and Jilin were assessed for their social development potential using eight tertiary indicators related to transportation infrastructure, traffic volume, and turnover. These indicators encompassed passenger and freight transportation, as well as the capacity of ports and airports. To facilitate radar chart analysis, each indicator was assigned a negative value, with higher values indicating higher development potential. Thus, a higher ranking meant a farther distance from the center of the circle. In general, this approach allowed for a comprehensive evaluation of the two regions and their respective social development prospects. Primorsky Krai's number of ports ranks third in the country. Its number of airports ranks eighth. These are the most important signs of its potential for social development. However, the performance of traffic and transit volume indicators does not match the volume. This indicates that Primorsky Krai's existing transportation infrastructure can provide significant support for regional development. This also indicates that Primorsky Krai's existing transportation infrastructure has potential for regional development. Jilin has similar results. In terms of transportation potential, the highest ranking indicator is "railroad mileage", which is ranked 15th in the country. Jilin is also ranked 19th in terms of the number of airports and ports. However, the performance in terms of traffic and transit volume does not match the volume, resulting in a low ranking for Jilin. This result shows that the existing transportation infrastructure of Jilin can provide a high level of support and potential for regional development. The poorly performing indicators in Primorsky Krai are road density (60th place) and passenger volume (53rd place). Due to its remote location, sparse population, and low passenger traffic volume, Primorsky Krai is relatively backwards in terms of road construction. However, the region has a unique advantage due to its cross-border nature. This unique advantage offers strong

potential for development. This potential could be realized if international cooperation is strengthened. The development potential of transportation in Jilin is similar to that of Primorsky Krai. Both regions benefit from relatively good transportation infrastructure and a unique advantage as cross-border areas, which hold strong potential for development with enhanced international cooperation. With such cooperation, the regions could leverage their existing strengths to further support regional growth and prosperity.

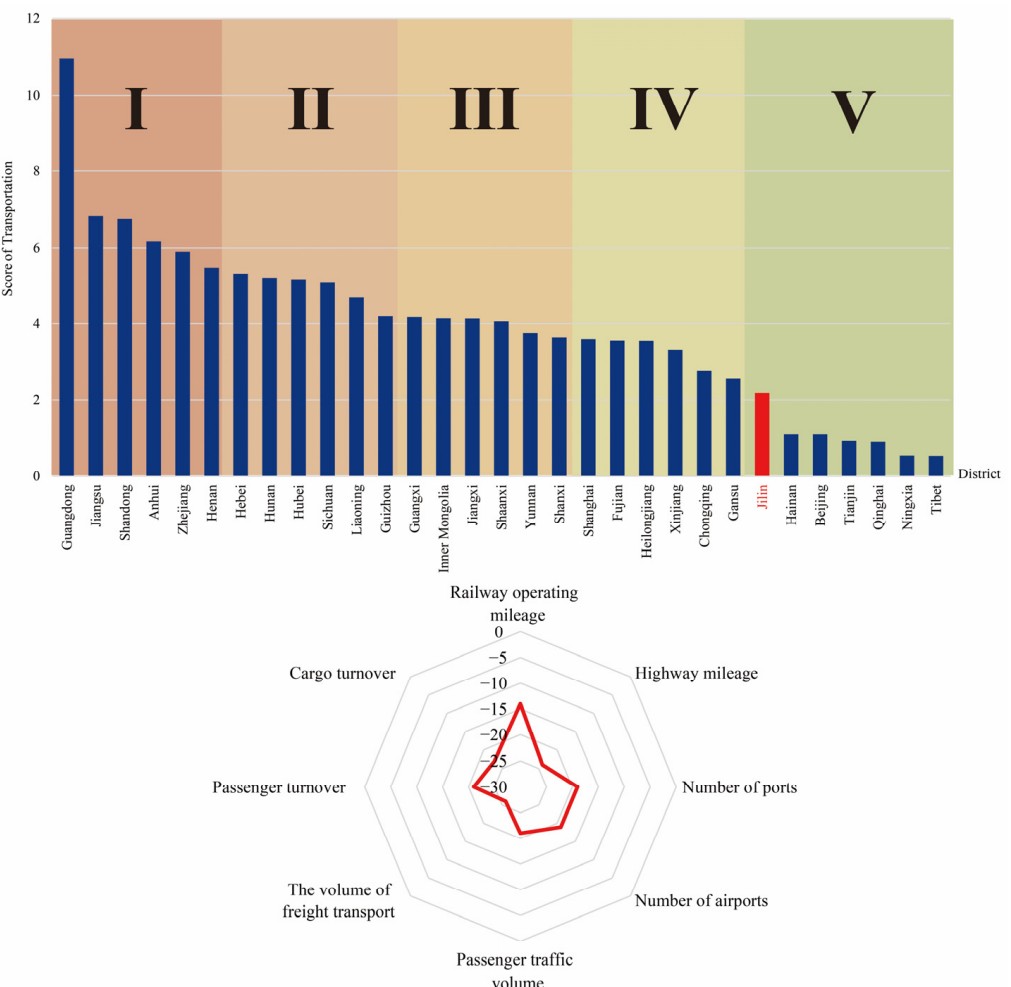

**Figure 8.** Jilin traffic development potential score and ranking of each indicator. (I—highest, II—high, III—medium, IV—low, and V—lowest).

Comparing the normative transport indicators of Primorsky Krai and Jilin, it is evident that the transport infrastructure of Primorsky Krai is far superior to that of Jilin, largely due to its abundant high-quality ports. Being a landlocked province, Jilin lacks maritime transport capacity and, therefore, needs to improve its foreign trade by increasing its reliance on maritime transport. Through cross-border cooperation in this area, both regions could achieve mutual benefits by leveraging their complementary advantages.

However, in terms of transport volume and turnover, Primorsky Krai lags behind Jilin. This suggests that the full economic potential of Primorsky Krai in transportation has not yet been realized, and cross-border cooperation in this area holds great promise for the region. With the right collaborative efforts and strategies, Primorsky Krai could maximize its untapped potential and further strengthen its economic position in the region.

### 3.4. Analysis of Resource and Environmental Potential

The resource and environmental development potential scores of each first-level administrative region in Russia and China were calculated individually using the re-

gional development potential assessment (RDPA) model, and the results are shown in Figures 9 and 10. Based on the RDPA scores, the researchers classified the 85 first-level administrative regions of Russia and 31 first-level administrative regions of China (excluding Taiwan Province, the Hong Kong Special Administrative Region, and the Macao Special Administrative Region) into five levels of very high, high, medium, low, and very low transportation development potential according to the quantile classification method, as indicated by different colors and labeled I, II, III, IV, and V, respectively, in Figures 9 and 10. The results show that Primorsky Krai (3.621) ranks 3rd among all 85 first-class administrative regions and belongs to the administrative regions with high resource and environmental development potential. Jilin (0.935) ranks 22nd and belongs to the administrative regions with low resource and environmental development potential.

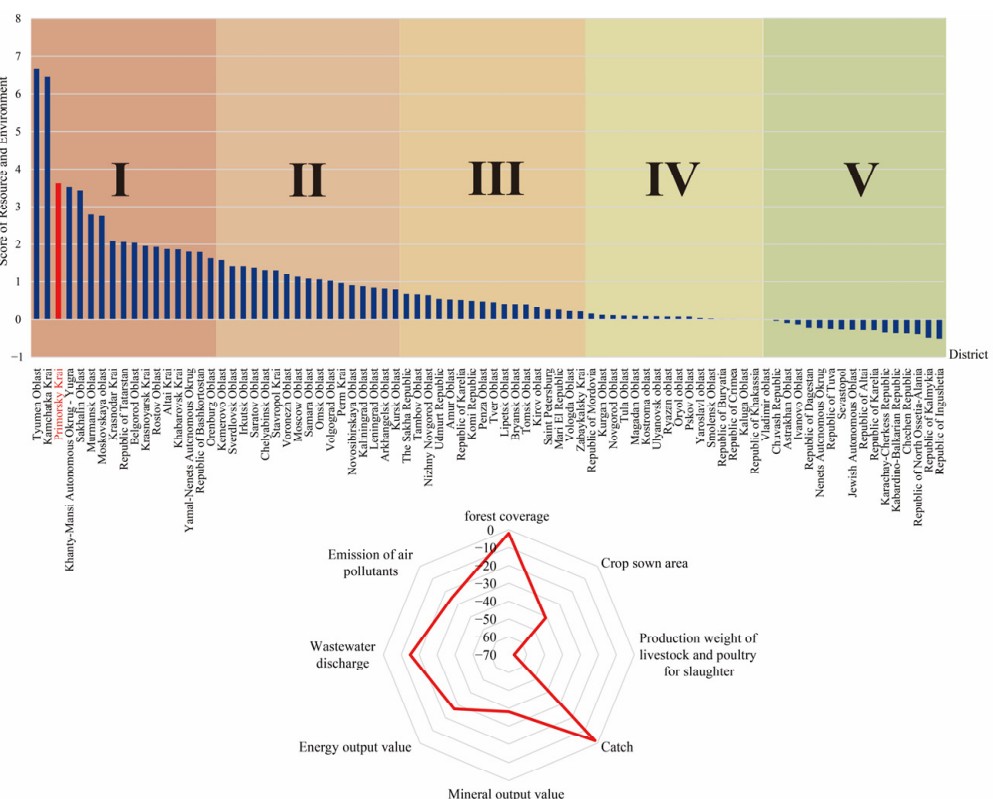

**Figure 9.** Primorsky Krai resource and environmental development potential score and ranking by indicators. (I—highest, II—high, III—medium, IV—low, and V—lowest).

Agriculture, forestry, animal husbandry, fishery, mining, wastewater, and waste gas were selected to assess the social development potential of Primorsky Krai and Jilin. To assist in examination of the radar chart, each indicator's rating was assigned a negative value; the farther from the center of the circle, the higher the ranking and the greater the potential for development. The highest ranking of Primorsky Krai in terms of the development potential of resources and environment is the second-highest in Russia in terms of fishing volume and forest cover, which represent the fishery resources. This is directly related to Primorsky Krai's geographical location, which is a mountainous coastal region. There is a strong potential for development if the relevant resource advantages are transformed into economic advantages and used wisely. Primorsky Krai sends out a little more wastewater and gas, so it needs better control and management of pollution. Primorsky Krai's development potential is limited by the poor performance of agriculture and animal husbandry as a whole. The region needs to work together to make up for this. The highest-ranking indicator of the resource and environmental development potential of Jilin is the total output value of the livestock industry, which is in 13th place. Jilin has rich forestry resources but a low forestry output value, which may be due to the

protection measures of closed mountains. In the same way, Primorsky Krai and Jilin have complementary relationships when it comes to their forests, minerals, fisheries, and agricultural resources. Their excellent resource endowments can help both regions, representing a potential win–win.

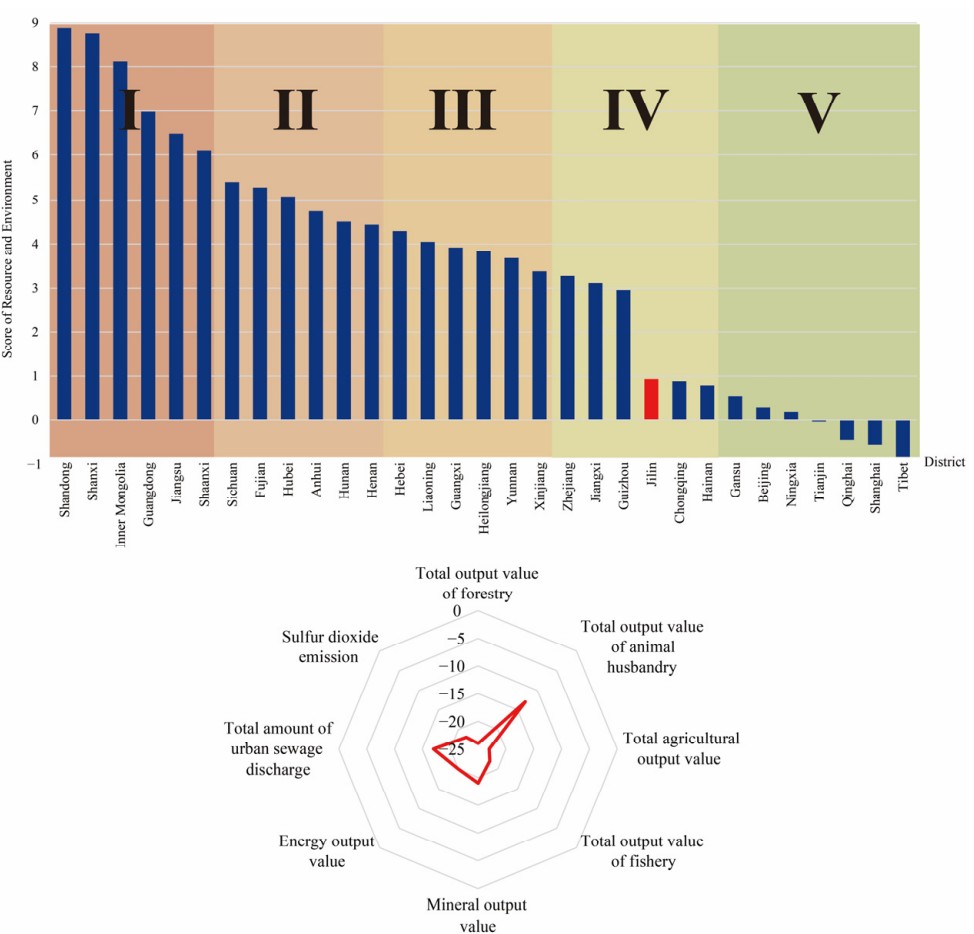

**Figure 10.** Jilin resource and environmental development potential score and ranking by indicators. (I—highest, II—high, III—medium, IV—low, and V—lowest).

Comparing the normative indicators of Primorsky Krai and Jilin at the transport level, it can be seen that the development potential of fishery resources, forestry resources, energy and mineral resources, and the environment in Primorsky Krai is higher than that of Jilin. Among them, the development potential of fishery resources in Primorsky Krai has a more significant advantage, while Jilin is weaker. In general, the natural resources of Primorsky Krai have obvious advantages over Jilin, and if one side can give full play to its resource advantages and the other side can give full play to its economic advantages, it will certainly produce better cooperation effects.

### 3.5. Comprehensive Evaluation of the Development Potential of Primorsky Krai

We calculated the composite development potential scores for each level of the administrative regions in Russia and China separately in this study. The results are shown in Figure 11. Based on the overall development potential scores, the researcher classified the 85 first-level administrative regions of Russia and 31 first-level administrative regions of China (expect for Taiwan Province, the Hong Kong Special Administrative Region, and the Macao Special Administrative Region) into five levels of higher, high, medium, low, and lower. These levels show the development potential according to the quantile classification method in Figure 11 by different colors and the labels I, II, III, IV, and V, respectively. The results show that Primorsky Krai (25.712) is ranked 11th out of all 85 first-class administra-

tive regions, which means that it has a high potential for overall development. Jilin (11.124) is ranked 26th among all 31 provincial administrative regions, which means that it has a low potential for overall development.

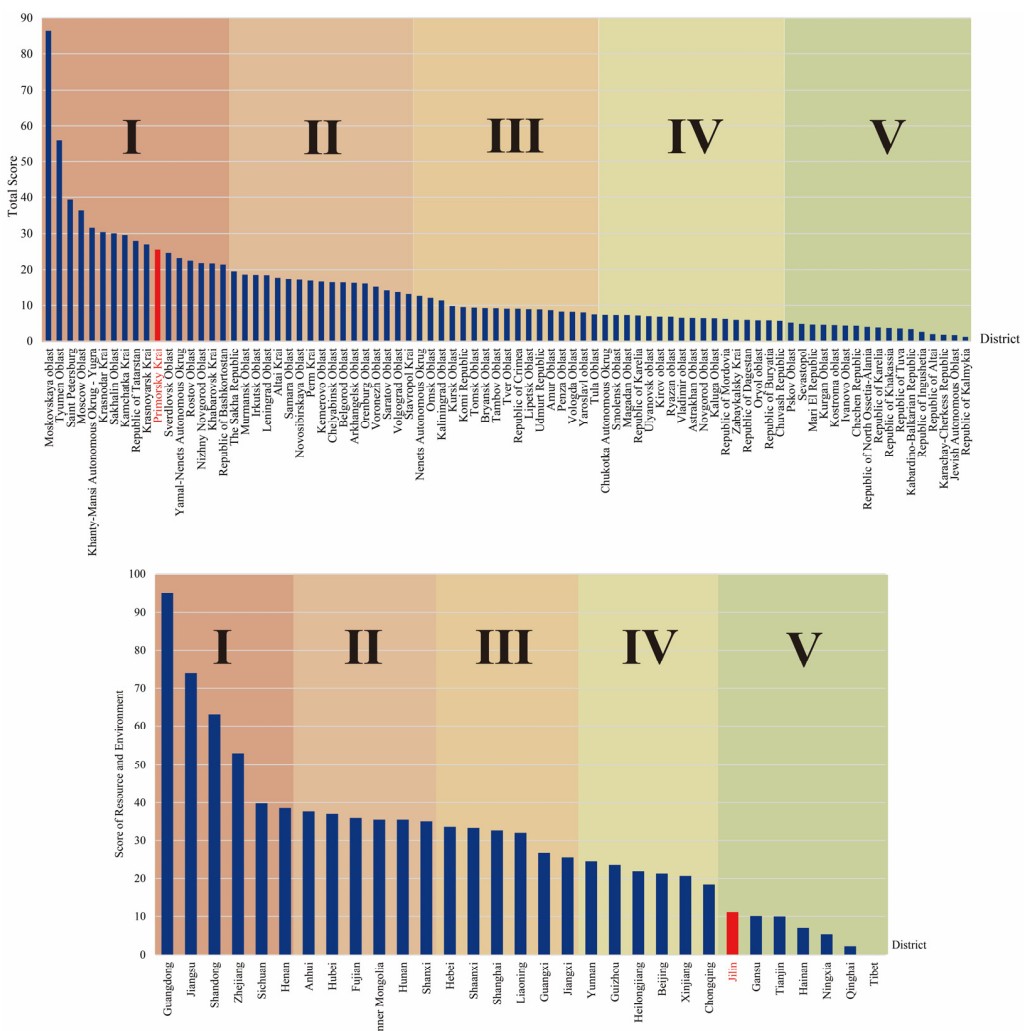

**Figure 11.** Development potential composite scores and grades. (I—highest, II—high, III—medium, IV—low, and V—lowest).

In terms of economic development potential, social development potential, transportation development potential, and resource and environmental development potential, Primorsky Krai ranks 29th, 31st, 15th, and 2nd, respectively. The region has the highest potential for resource and environmental development, the second-highest potential for transportation development, and a slightly lower potential for economic and social development. However, it is still in the more developed part of all first-class administrative regions. Primorsky Krai has growth potential if it takes advantage of its location in a cross-border zone and works with other states. Jilin ranks 25th in terms of economic development potential, 26th in terms of social development potential, 25th in terms of transportation development potential, and 22nd in terms of resource and environmental development potential. Jilin's overall development potential is low, but the region has distinct advantages in cross-border logistics and trade, agricultural development, talent training, and industrial development. Furthermore, because Jilin shares a border with Primorsky Krai, it can use their geographical characteristics to complement one another and collaborate closely for future joint development.

## 4. Cross-Border Cooperation Direction

1.  *Primorsky Krai Has a Thriving Logistics Industry and a High Demand for Infrastructure Upgrades. Jilin Lacks Port Facilities. China's Experience with Global Infrastructure Construction Is Mature. Cooperation in Cross-Border Transportation Infrastructure Projects between China and Russia is Critical.*

According to the preceding model, Primorsky Krai ranks 15th in terms of transportation, which is extremely high. "Transportation infrastructure" is an important indicator. Primorsky Krai ranks third in port numbers in the country, which gives it a strong regional advantage in port logistics. However, the infrastructure of roads (ranked 60th) and railroads (ranked 56th) is relatively weaker than the developed port facilities. Russia has a big problem with infrastructure that is old and not being updated quickly enough. For example, the port of Vladivostok has trouble bringing in and sending out Chinese cargo because it does not have enough space. The berth is in the middle of the city, and there is no road that can be used to bring goods in and out the port. This has caused goods waiting to be piled up. Primorsky Krai also ranks higher when it comes to the volume (46th) and turnover (48th). There is also a problem with the existing infrastructure. Thus, it seems that as cross-border cooperation grows in Primorsky Krai, there will be a strong demand for infrastructure construction.

The lack of seaports is currently the most serious transportation issue confronting Jilin Province. This issue has a significant negative impact on Jilin Province's transportation capacity. This issue can also be derived from the indicator of "transportation development level" (22nd). Every year, tens of thousands of bulk products arrive in Jilin Province—China's heavy industry and commodity grain hub. However, the lack of its own seaport makes transporting bulk products from the north to the south particularly challenging. Jilin's goods, for example, must first be carried by train to Liaoning before being shipped across the Bohai Sea. If Jilin's commodities are to be exported to Europe, the United States, or Japan, they must pass via the Korean Strait and into the Sea of Japan. The distance between Jilin Province and the Sea of Japan is only 15 km. High transportation costs have slowed import and export trade as well as the economic development in Jilin Province.

By combining Jilin's disadvantages, such as a lack of ports, with the advantages of Primorsky Krai, we can achieve a win–win situation for both regions. Furthermore, China has had many years of development and construction. China has rich experience and technology in domestic and international transportation infrastructure construction. Primorsky Krai and China can collaborate to build transportation infrastructure in Primorsky Krai for a joint growth of both regions.

At present, Primorsky Krai's port logistics industry is responsible for 80% of maritime transportation in the Far East. Vladivostok, Nakhodka, and Orient ports are the three most famous seaports in Primorsky Krai. They are deep-water, non-freezing ports. From what we already know, cooperation between the ports of Jilin and Primorsky Krai has more benefits. The Hunchun–Zarubino Port–Ningbo Zhoushan Port cross-border route for domestic cargo opened in 2018. It is 300 to 800 km shorter than the existing route through Dalian and Yingkou ports from Jilin, and it takes 2 to 4 days less. Accordingly, RMB 10–20 can be saved per ton on transportation costs. The cross-border traffic routes between China and Russia are still relatively rare. It is urgent to expand the construction of cross-border traffic facilities to promote further leapfrog development of international trade. The Jilin and Binhai border regions should work together in the future based on international transportation, logistics, and infrastructure. This will directly drive the growth of the economies in the Russian Far East and Northeast China. It will also encourage the win–win development of the port and the surrounding area, with promising potential.

2.  *The Coastal Krai Is Rich in Agricultural and Fishery Resources, and the Cooperation between Jilin and the Coastal Krai in Agriculture and Other Large-Scale Planting, Aquaculture Technology, and Trade Is Promising.*

According to the model, the agricultural resources of Primorsky Krai are ranked in 41st place in the country. This was calculated based on the area of sowed crops. Primorsky Krai has 1,649,400 ha of agricultural land, accounting for 10% of the region's total geographical area. Primorsky Krai has a small population but a vast amount of undeveloped territory. As a result, Primorsky Krai has a reasonably large supply of agricultural resources to provide.

Jilin is also of agricultural importance in China, with reasonably mature mechanized agricultural production expertise. In addition, over the last five years, we computed the import and export of agricultural products in the immediate vicinity. The specific results are shown in Table 2. We discovered that agricultural commodity imports in Jilin Province have achieved fresh highs since 2018. This indicates that agricultural products are in high demand in Jilin Province. Primorsky Krai can therefore take advantage of the potential to expand its agricultural production on a vast scale. We believe that Primorsky Krai and Jilin may collaborate in farming and planting technology for joint development.

**Table 2.** The 2016–2020 import and export volume of agricultural products in Jilin.

| Unit | Agricultural Exports | Import of Agricultural Products | Total Trade | Import and Export Balance |
|---|---|---|---|---|
| | Billion CNY | Billion CNY | Billion CNY | Billion CNY |
| 2016 | 1.32 | 6.09 | 7.41 | −4.77 |
| 2017 | 0.83 | 8.21 | 9.04 | −7.38 |
| 2018 | 0.72 | 21.19 | 21.91 | −20.47 |
| 2019 | 0.49 | 17.38 | 17.87 | −16.89 |
| 2020 | 0.44 | 19.15 | 19.59 | −18.71 |

Primorsky Krai's agriculture is centered in the south and southwest, with 160,000 ha of agricultural area and an average total agricultural output of RUB 7.15 billion. Soybeans and potatoes are among the most important crops farmed. COFCO Far East Ltd.—China's largest state-owned grain, oil, and food production corporation in the region—has made an investment in Vladivostok. It regularly participates in large-scale soybean and corn trade, importing 19,000 T of Russian soybeans in 2019. There are promising opportunities for cooperation between the two parties in the development of grain storage and logistical systems. There is currently a shortage of vegetables and other agricultural products in the Russian Far East, and special agricultural products in Jilin such as ginseng, licorice, black fungus, edible mushrooms, Chinese wolfberry, and blueberries are welcomed by the Russian Far East market and have high consumer demand. Jilin and Primorsky Krai also have good opportunities to trade agricultural technology.

Based on the previous section in the model, Primorsky Krai has great fishing resources, where it ranks in second place in the country. The coastal Krai has a lot of places to fish. Primorsky Krai is one of the main places in Russia to obtain fresh fish, canned fish, and other kinds of seafood. It produces about 1 million tons of fish and seafood every year, and more than 50,000 people work in fishery-related jobs there. Vladivostok and Nakhodka both have special fish ports. According to information from the All-Russian Association of Fishery Enterprises, Primorsky Krai exported more fish products than any other part of the Far East. The amount was USD 1.746 billion in 2022, which was 1.7% more than the year before. Most of the fish that Russia exports from the Far East goes to Asia. South Korea, China, and Japan are the most important export markets.

Aquaculture cooperation between Russia and China plays an essential role in Russian–Chinese trade. The two countries' cooperation in this field has enormous potential for development. However, Primorsky Krai is currently dealing with some issues in the aquaculture business. First and foremost, the marine fisheries are dominated by fishing, with no large-scale industrial cultivation. China's extensive aquaculture experience can provide technological assistance in the transformation and upgrading of the aquaculture industry in Primorsky Krai. Jilin Hunchun works as a sub-center of the Sino–Russian

fisheries platform in the Northeast Asia International Fisheries Joint Laboratory project. It represents a fresh approach to Sino–Russian technical commercial collaboration. At the same time, the two countries are working together to breed juvenile shellfish in Nakhodka.

Furthermore, exorbitant transportation expenses do not result in a price advantage for Far Eastern fishing products on the domestic and foreign markets in Russia. This is a significant element that hinders the growth of Far Eastern fisheries. Due to the rising demand, China's per capita consumption of main fish products has been rising since 2014 (Figure 12). Fish goods are in high demand in Jilin. A total of CNY 1.442 billion in fish products were imported from Russia in 2020. Jilin Province is located next to Primorsky Krai. Primorsky Krai has abundant fish resources, and there is a high demand for fish products in Jilin Province. Cooperation in fish trade could lessen the impact of freight costs on prices and increase both sides' profitability. Furthermore, Jilin is surrounded by the vast Chinese market, and if Primorsky Krai can use Jilin as a node to better harness its economic and trade potential in China, its economic development will improve.

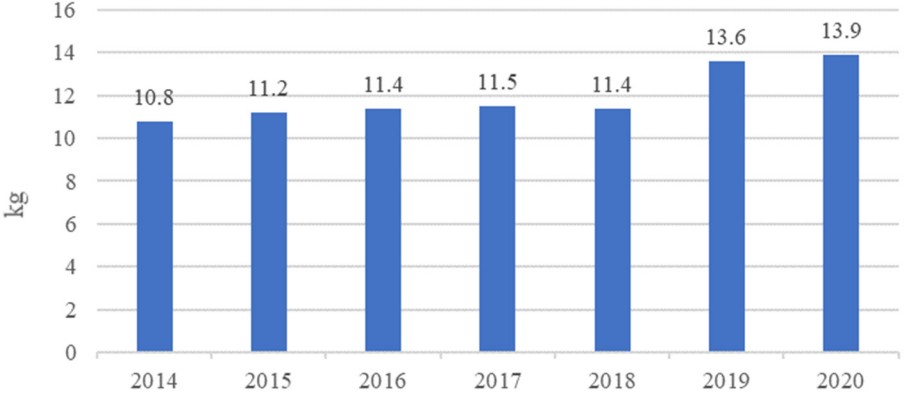

**Figure 12.** Per capita consumption of major fish products in China (kg).

3.  *Primorsky Krai Is Rich in Mineral Resources, and the Two Sides Can Carry Out In-Depth Cooperation in the Field of Mineral Resources Development and Deep Processing*.

Primorsky Krai is one of the most abundant areas of energy and minerals in the Far East. This is due to its unique geographical location and geological conditions. The model indicates that Primorsky Krai ranks 38th in the nation in terms of the value of its mineral production. Most of the mineral resources have not been fully exploited. Therefore, Primorsky Krai has a high potential for mineral resource development.

Primorsky Krai is rich in mineral resources, and the most important part of its economy is the mining and processing of non-ferrous metals. Primorsky Krai is where 80% of the fluorspar in Russia comes from. Hydrofluoric acid can be made from fluorspar, which can be used to make cryolite for the aluminum industry. Russia has the third-largest tin mines in the world. Primorsky Krai has about 30 places where tin is mined, mostly in Kavaloyarov, Far Side, and Krasnoyarsk. About 50 gold mines are in Primorsky Krai, most of which are in the south and north. Primorsky Krai has a strong advantage in working with other countries because of its abundance in mineral resources.

According to the model, the mineral resources of Jilin Province are ranked 19th, which is in the median level. Notably, the reserves-to-mining ratio for Jilin's primary minerals—such as coal, iron, copper, nickel, lead, zinc, gold, boron, and others—has been dropping for years, and most mines have entered their middle and late stages. In addition, there is no resource succession within the Bureau of Mines, resulting in a significant demand gap for the most important minerals. Jilin must seek cooperation from others to solve this situation. Similarly, this holds true for early resource development. Jilin has higher expertise in mineral resources development. It is highly capable of deep processing. Therefore, technology is an advantage of the mining sector in Jilin for international collaboration.

Previous research indicates that Jilin and Primorsky Krai have a promising future in mining and processing. Currently, the Jilin Hunchun Comprehensive Free Trade Zone is focused on the development of the "overseas mining, domestic processing" model for copper, gold, tin, and other mineral goods. This enables the two locations to cooperate in the sector of mineral resources, laying a solid platform for future collaboration development.

4.  *Both Sides Have a Promising Future in Talent Training and Cross-Border Labor Cooperation.*

The most active component of production is population. This is the primary factor in converting raw materials into social prosperity. Jilin and Primorsky Krai are both experiencing more challenging population issues. The population of Primorsky Krai and Jilin is shown in Table 3.

**Table 3.** Population of Jilin and Primorsky Krai.

|  | Jilin | Primorsky Krai |
| --- | --- | --- |
| Total population (10,000 people) | 2375.37 | 189.6 |
| Population density (person/km$^2$) | 1447.51 | 11.7 |
| Net emigration of population (10,000 people) | 23.97 | 2791 |
| The proportion of the total population (‰) | 10.1 | 1.4 |

Primorsky Krai scores only 50th in population growth, 43rd in population indicators at the guideline level, and 37th in migration increase per 10 million people. The Far East has experienced declining population growth in recent years. Due to this severe population decline, the Far East's ability to extract energy, manufacture goods, and engage in other economic activities has been constrained. The economic growth of the Far East has been impacted. We can see that Primorsky Krai's demographic condition is not promising. The Russian government has implemented a number of beneficial policies to boost regional birth rates, stop population exodus, and draw foreign migrants to meet the labor needs in the Far East, including encouraging migration and rewarding delivery. Primorsky Krai's present labor shortage issue, however, has not significantly improved. Therefore, increasing the number of foreign laborers might be a solution.

Additionally, Jilin, which is ranked 31st in the model and at the bottom of the nation, has a more serious population issue. With a net migration of 239,700 people in 2020, Jilin is suffering from demographic problems due to its low natural growth rate and high population outflow. The issue of population decline in Jilin is partially brought about by the province's reduced employment opportunities.

Primorsky Krai has a high demand for cross-border labor cooperation. Jilin has a sizable population and a plentiful labor force. Cooperation on the labor market would considerably alleviate Primorsky Krai's manpower shortage for joint development.

The labor force collaboration and exchange makes it equally important to interchange scientific and technological talent in both directions. According to our findings, Jilin (ranked 22nd) and Primorsky Krai (ranked 25th) both have solid foundations for future growth in the fields of science and education. They also have higher potential for growth. By depending on the exchange of knowledge and technology, the two areas may depend on one another and advance together.

Primorsky Krai and Jilin have a lot of potential for cross-border talent exchange and in-depth science and technology collaboration. Primorsky Krai is a cultural, educational, and scientific research center in Russia's Far East, with over 20 institutes of higher learning in its territory. With 21 postgraduate training institutions and 62 general colleges and universities, Jilin is also rich in educational resources. Jilin University, Yanbian University, and other Chinese universities are currently collaborating with Russian universities. Talent is crucial for long-term growth for both sides. To expedite win–win economic cooperation between Jilin and Primorsky Krai, China and Russia should put more emphasis on talent training and introduction and make it a priority in economic collaboration.

China and Russia celebrated the "Year of Science and Technology Innovation" in 2020 and 2021. Talent exchange across the border between Jilin and Primorsky Krai is important for the cooperation of China and Russia in science and technology innovation. It is very important for China and Russia to work together on scientific and technological innovation and intellectual exchange, to make the achievements of cooperative education reach more people and train more talent.

## 5. Conclusions

Based on the above discussion, we can draw the following conclusions:

First, cooperation between Primorsky Krai and Jilin across the border has a lot of potential. At the moment, research shows that both Russia and China want to work together across the border in Primorsky Krai and Jilin. China's "One Belt, One Road" initiative and Russia's "Look East" strategies seem to fit well. This is an important policy basis for cross-border cooperation between the two countries. China and Russia's trade continues to grow, and closer economic and trade cooperation has been set up. This gives this cross-border cooperation a strong economic foundation. Though geopolitical factors that cause the economic and trade conflict are ongoing between the East and West, China and Russia have the potential to collaborate more, and it seems to be the right time. Meanwhile, Russia's plan for developing the Far East fits with China's plan for reviving the Northeast. Other plans for regional cooperation that are consistent will provide guidelines for joint efforts across borders. Regional cooperation between China and Russia is based on the fact that both countries have shared interests. Even though there are differences in many areas, a Sino–Russian community of shared interest could help the two countries work together even closer.

Second, the proposed cross-border cooperation between Primorsky Krai and Jilin can be characterized in four directions: First, international transportation, logistics, and infrastructure cooperation in the border area would immediately stimulate economic growth and the flow of resources between the two regions, as well as promoting the joint development of the ports and the hinterland. Second, the two regions have a substantial demand for agricultural and marine products. Possibilities for cooperation between the two locations exist for the exchange of pertinent technologies to meet the need for out-of-season produce and fisheries resources. Third, the two locations could concentrate on the growth of mineral resource development and processing in the Jilin Hunchun Comprehensive Free Trade Zone in order to establish a "coastal frontier mining, Jilin processing" kind of mineral resources deep processing business. To boost economic growth in the Far East and Jilin Province, the two areas could also collaborate in higher education, science, and technology, as well as talent training and labor collaboration.

Furthermore, the investigation of the development potential of Primorsky Krai–Jilin cross-border cooperation demonstrates that the development of cross-border regional cooperation is a crucial research topic in the context of expanding the global division of labor and cooperation. The existing analysis of a single border region cannot adequately support regional development; therefore, it must be combined with a synergistic analysis of the corresponding region of another country across the border in order to assess the overall development potential of the region more scientifically and adequately. This study conducted a synergistic analysis of cross-border regions of Russia and China, and a more scientific assessment would be strengthened by thoroughly considering the counterparts of both sides on the basis of cross-border cooperation studies.

## 6. Discussion

### 6.1. This Study Emphasizes the Significance of Cross-Border Research

We broaden the cross-border analysis of a particular region to include the equivalent region. These original border regions have significant economic and trade linkages to their neighboring regions. This cross-border regional analysis is more realistic and comprehensive. The direction of this study is distinct from other border investigations.

Previous studies mostly concentrated on border regions' own economic development concerns [51,52] or finance [53,54], and other single areas of associated collaboration. We treat regions on both sides of the border as one object of research. We are interested in the complementarity of cross-border zones. As a result, we find new opportunities for cross-border economic cooperation. We believe that conducting cooperation research on a group of cross-border regions is more meaningful than conducting cooperative studies on specific border locations. After all, cross-border zones are always there. We cannot afford to ignore either side.

### 6.2. Cross-Border Cooperation Research Is Challenging

Field research is regarded as quite significant in this study. A significant amount of fieldwork was conducted. In our study, first-hand data from our fieldwork were more significant. Interviews with representatives of the government, development strategy specialists, and local people can provide us with more pertinent development knowledge and identify the real-world challenges. In addition, owing to the different statistical standards, field research might help to lessen the mistakes brought by these variations. A thorough scientific investigation is more pertinent in the study area than a simple statistical examination of the data. We believe that this study's findings are more scientifically valid.

### 6.3. The Development Potential of Cross-Border Cooperation Hot Zones Was Assessed

Our findings offer solid scientific backing for practical development. However, issues with the fundamental guidelines for cross-border collaboration must be resolved. In order to be more effective, we should encourage cross-border cooperation in regions of high geo-economic development importance. For future research, we must address ensuing concerns, such as the following:

- Investigating the theories governing cross-border economic cooperation and identifying those that have broad relevance.
- Tracking research on areas with cross-border collaboration to continuously improve the methodology for potential analysis.
- Checking whether there are more sophisticated models of cross-border collaboration that actually stabilize, secure, and benefit both parties.

Of course, we need to conduct more investigations to advance the cross-border regional economic geography studies together.

**Author Contributions:** Conceptualization, M.J. and F.L.; Data Curation, M.J., Y.Z., A.B. and S.X.; Formal Analysis, M.J., Y.Z., K.G. and F.L.; Funding Acquisition, F.L.; Investigation, M.J., Y.Z., A.B. and S.X.; Methodology, F.L.; Resources, T.B.; Supervision, T.B., Y.Z. and F.L.; Writing—Original Draft, M.J.; Writing—Review and Editing, M.J., F.L., S.X., Y.Z., T.B., A.B. and K.G. All authors have read and agreed to the published version of the manuscript.

**Funding:** National Science and Technology Basic Resources Investigation Project of China, funded by the Ministry of Science and Technology of China (Grants No. 2022FY101903).

**Institutional Review Board Statement:** Not applicable.

**Informed Consent Statement:** Not applicable.

**Data Availability Statement:** Some or all data or models that support the findings of this study are available from the corresponding author upon reasonable request.

**Acknowledgments:** The authors would like to thank the editor and reviewers for their insightful comments and suggestions.

**Conflicts of Interest:** The authors declare no conflict of interest.

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
