# Peer review of "Potential for Economic Transition and Key Directions of Cross-Border Cooperation between Primorsky Krai (Russia) and Jilin (China)"

_sustainability, doi:10.3390/su15054163_

Round 1

Reviewer 1 Report

The introduction section needs to be more focused on the topic of the article, rejecting a broad and sweeping narrative. The section needs to focus on the evolution and development related to the potential for economic transformation. The literature currently cited is sparse and lacks up-to-date literature as well as key highlights. More literature needs to be added.

The analysis of the findings is still relatively simple and needs to be further explored. It is recommended that the article adds innovative research methods to further dissect the results and present conclusions.

Author Response

We are very grateful for your comments regarding our manuscript. All your suggestions are very important to us, both for composing the manuscript and our further research. We have studied comments carefully and have made corrections which we hope meet with approval.

Point 1: The introduction section needs to be more focused on the topic of the article, rejecting a broad and sweeping narrative. The section needs to focus on the evolution and development related to the potential for economic transformation. The literature currently cited is sparse and lacks up-to-date literature as well as key highlights. More literature needs to be added.

Response 1: Thank you very much for the suggestions on supplementing the introduction. We have revised the manuscript.

According to your suggestion that the introduction should pay more attention to the topic of the article, we have revised the current research situation and deficiencies in the introduction as a whole, added the review of the current research on the potential of economic transformation, and paid attention to the timeliness of the literature. It made the content more accurate and fit the research in this paper.

This paper focused on the development potential of cross-border regional cooperation, so it reviewed the relevant literature of existing research. Generally speaking, cross-border cooperation is one of the forms of territorial cooperation between different types of units in border areas. The common historical origins and other important similarities between neighboring regions on the border, and to achieve common development goals. Cross-border cooperation is a recognized way to develop the economy of border areas.

The existing studies on specific cross-border areas are mostly concentrated among European countries. China and Russia have strong differences in economic development and have a good will to cooperate in cross-border economic cooperation. This determines the high research value of cross-border economic cooperation between Primorsky Krai and Jilin. In addition, we also added the review of relevant research on the potential of economic transformation, introduced the latest concept of sustainable economic transformation, and sorted out the meaning and development of "transformation geography". The study of transformation geography paid less attention to developing countries, focused on single case analysis, and lacked regional comparative analysis and exploration. Primorsky Krai and Jilin are in line with the typical areas of sustainable economic transformation and have high research value.

Through the above literature review and analysis, we summarized two deficiencies of the existing research. First, there was a lack of key cross-border areas in the Far East and Northeast China and Russia. The cross-border cooperation between Primorsky Krai and Jilin is crucial to promote the construction of the China-Mongolia-Russia Economic Corridor and the regions along the line development. Especially under the situation of intensified Russian-Uzbekistan conflict, good cross-border cooperation between the two places is significant for promoting the stability of the development environment in the Eurasian continent. Second, at present, most of the studies are on one country and one region, with a single region as the entry object, and lack comprehensive comparative analysis. The study on the development potential of cross-border regional cooperation between Primorsky Krai and Jilin could well supplement.

Point 2: The analysis of the findings is still relatively simple and needs to be further explored. It is recommended that the article adds innovative research methods to further dissect the results and present conclusions.

Response 2: Thank you very much for the suggestions on research results and methods. The relevant parts of the original manuscript were indeed not explained in detail. After the team discussion and in-depth analysis, it is revised as follows:

In this study, we build an evaluation index system of regional development potential from the perspective of cross-border cooperation. We sorted out the relevant data of the field survey of the Primorsky Krai and Jilin, including the Russian Statistical Yearbook 2020, the Development Outline of the Russian Far East Border Region 2015-2025, and the data of the symposium with the Primorsky Krai Government and the Jilin Provincial Government. According to the content of the data, and the location characteristics of the two places, we established four subsystems including economy, society, transportation, and resources and environment. The indicators of each subsystem included all aspects of social and economic development and are closely related to cross-border economic cooperation. In the process of determining the indicators, we consulted the opinions of many experts and tried to be scientific. At the same time, we also took into account the differences in the statistical caliber of the two countries, so that similar indicators could correspond as much as possible. Finally, we established a multi-level regional development potential assessment (RDPA) of the target layer, element layer and indicator layer. In the 2.2.2 statistical methods section, we added the comparative analysis of the development potential between Primorsky Krai and Jilin to make the model more complete. Through the calculation in the previous steps, we could get the scores of the indicators of Primorsky Krai and Jilin. Here, we introduced a comparative analysis model to make the analysis results clearer. Similarly, in the third part, we have added an explanation of the comparison results. The relevant indicators of the four subsystems are involved and further analyze the relevant results. In summary, we have two innovations in the methodology of this study. First, the innovation of indicator systems and data. This paper is an integrated analysis of Russia and China's economy, society, transportation, resources and environment. The data comes from the cross-border field survey of the Ministry of Science and Technology project, including data collection, government seminars, expert interviews, etc., which is unique. Second, this research model is a comparative and comprehensive analysis of different countries and regions. Such research is different from single regional research and has the characteristics of complexity, contrast and cross-border. In this study of cross-border cooperation, "different countries" are Russia and China, and "different regions" are the Primorsky Krai and Jilin, which are geographically adjacent but have obvious differences. Through comparative research, we could explore and summarize the differences and turn them into the direction of cross-border cooperation, which is an important innovation.

We have added 5. Conclusions. After using the above model method for data analysis, we conducted further discussion in this section. First, from the whole research process, we could easily conclude that the potential for cross-border cooperation between Primorsky Krai and Jilin is huge. This potential for cooperation is not only reflected in the geographical proximity of the two countries, but also reflected in many aspects such as the economic strategic alignment of the two countries, the sustained growth of bilateral trade, and the strengthening of cooperation would due to the trend of the world situation. At present, research on cross-border cooperation has paid little attention to the two regions of the Primorsky Krai - Jilin. The detailed carding of this study has made it clear why the research area is important. Second, what is the future cooperation direction of the cross-border cooperation between Primorsky Krai and Jilin, and how should the two sides carry out cooperation to achieve win-win results? We believe that the cross-border cooperation between Primorsky Krai and Jilin has a good development prospect in terms of port logistics infrastructure cooperation, greenhouse agriculture and marine fishery trade cooperation, mineral resources development and deep processing cooperation, talent training and cross-border labor cooperation, and relevant cooperation directions are also described in the manuscript. Thirdly, in terms of the research methods of cross-border cooperation, through the case study, we believed that the development of cross-border regional cooperation needs to be combined with the collaborative analysis of another country adjacent to its cross-border border. For example, to analyze the cross-border cooperation in the Primorsky Krai, it is not convincing to discuss the relevant content of cross-border cooperation only based on an independent analysis of the region. Who should cooperate with and how to cooperate with the Primorsky Krai are all outstanding issues. At this time, Jilin, the cross-border adjacent area of the Primorsky Krai be proposed separately. We considered the situation of the two regions fully when establishing the model so that the cross-border collaborative analysis results were more scientific, targeted, and practical. Fourth, this study could provide scientific support for promoting the two countries to strengthen cross-border cooperation and efficiently promote the construction and rapid development of economic corridors. In the context of globalization, exchanges and cooperation between countries have become increasingly frequent, which has created a new demand for the evaluation of cross-border regional development potential. Our research could solve the problems related to cross-border regional cooperation from a more scientific perspective. This will also have a profound impact on the cross-border cooperation between the two countries and the regional economic construction.

Reviewer 2 Report

After reviewing the document, it is considered that it should be rejected, based on the following arguments:

First, the literature review is deficient, there is no evidence of a theoretical approach on which to support, argue and explain the work. In its current state, the writing is more similar to a technical report than to a research.

Secondly, the method proposed by the researchers has similar characteristics of a widely studied method such as Ordered Weighted Average OWA proposed by Yager in 1988, as well as the estimation of the weights found in Yager's studies of 1988 and 2002. In this sense, it is important that the authors make a better review of the method and other methods that may coincide and give the necessary credits so as not to fall into plagiarism problems. The potential of the database argued by the authors is highlighted, which can be used to its full potential.

Thirdly, in all research it is expected that there is a defined discussion of the results found to contrast with the theoretical and give more solid statements. This problem stems from the insufficient literature review. Likewise, the document lacks the conclusions where those relevant aspects of the work are consigned as well as its implications, limitations and future lines of research.

Author Response

We are very grateful for your comments regarding our manuscript. All your suggestions are very important to us, both for composing the manuscript and our further research. We have studied comments carefully and have made corrections which we hope meet with approval.

Point 1: First, the literature review is deficient, there is no evidence of a theoretical approach on which to support, argue and explain the work. In its current state, the writing is more similar to a technical report than to a research.

Response 1: Thank you for the comments on the literature review. After extensive study and research, we have fully revised the literature review part of the manuscript. Based on your comments, we have revised the literature review part of the article as a whole, adding theoretical methods to support, demonstrate and explain relevant research reviews, to enhance the academic nature of this article. The literature review focused on the development potential of cross-border regional cooperation. We sorted out the concept and significance of cross-border cooperation, and listed the study of Polish - Lithuanian border and the sustainable cross-border cooperation of the Czech - Polish border. In addition, we added the review of relevant research on the potential of economic transformation, introduced the latest concept of sustainable economic transformation, and sorted out the meaning and development of "transformation geography". The essence of regional economic transformation is a process of institutional change and institutional innovation, which provides a new incentive structure for improving the quality of economic growth and internalizes external effects.

Cross-border regional cooperation includes the idea of "transformation geography". It focused on the multi-scale geographical co-performance of emerging transformation and development, as well as the dynamics of power relations and regional unbalanced development behind it, and has become an important research content of "evolutionary economic geography". It also could provide theoretical support for this study. The study of transformation geography paid less attention to developing countries. At present, the existing research focuses on single-case analysis and lacks regional comparative analysis and exploration. Primorsky Krai and Jilin are in line with the typical areas of sustainable economic transformation and have high research value.

Through the above literature review and analysis, we summarized two deficiencies of the existing research. First, there was a lack of research on the key cross-border areas in the Far East and Northeast China and Russia. The cross-border cooperation between Primorsky Krai and Jilin is crucial to promote the construction of the China-Mongolia-Russia Economic Corridor and the development of the regions along the line. Especially in the situation of the intensification of the Russian-Uzbekistan conflict, good cross-border economic and trade cooperation between the two places is of greater significance to promote the stability of the development environment in the Eurasian continent. Second, at present, most of the studies are on one country and one region, with a single region as the entry object, and lack comprehensive comparative analysis. The study on the development potential of cross-border regional cooperation between Primorsky Krai and Jilin could well supplement. This study would make a comparative study of the key areas of cross-border adjacent cooperation, enrich the research methods for evaluating the development potential of cross-border regions, and provide scientific support. The Primorsky Krai -Jilin Cross-border Cooperation Zone has strong academic research value.

Point 2: Secondly, the method proposed by the researchers has similar characteristics of a widely studied method such as Ordered Weighted Average OWA proposed by Yager in 1988, as well as the estimation of the weights found in Yager's studies of 1988 and 2002. In this sense, it is important that the authors make a better review of the method and other methods that may coincide and give the necessary credits so as not to fall into plagiarism problems. The potential of the database argued by the authors is highlighted, which can be used to its full potential.

Response 2: Thank you very much for the comments. According to the requirements of the reviewers, we have made corresponding modifications to this part.

The research method is an important part of the research, and we have sorted out and reviewed the model at your suggestion. We believe that there are two innovations in the methodological part of this study. First, the innovation of indicator systems and data. This paper is an integrated analysis of Russia and China's economy, society, transportation, resources and environment. The data comes from the cross-border field survey of the Ministry of Science and Technology project, including data collection, government seminars, expert interviews, etc., which is unique. Second, this research model is a comparative and comprehensive analysis of different countries and regions. Such research is different from single regional research and has the characteristics of complexity, contrast and cross-border. In this study of cross-border cooperation, "different countries" are Russia and China, and "different regions" are the Primorsky Krai and Jilin, which are geographically adjacent but have obvious differences.

In this study, we build an evaluation index system of regional development potential from the perspective of cross-border cooperation. We sorted out the relevant data of the field survey of the Primorsky Krai and Jilin , including the Russian Statistical Yearbook 2020, the Development Outline of the Russian Far East Border Region 2015-2025, and the data of the symposium with the Primorsky Krai Government and the Jilin Provincial Government. According to the content of the data, the location characteristics of the two places and the research direction of cross-border cooperation, we established four subsystems including economy, society, transportation and resources and environment. The indicators of each subsystem include all aspects of social and economic development and are closely related to cross-border economic cooperation. In the process of determining the indicators, we consulted the opinions of many experts and tried to be scientific. At the same time, we also took into account the differences in the statistical caliber of the two countries, so that similar indicators can correspond as much as possible. Finally, we established a multi-level regional development potential assessment (RDPA) of the target layer, element layer and indicator layer. In the 2.2.2 statistical methods section, we added the comparative analysis of the development potential of the cross-border regional cooperation between Primorsky Krai and Jilin to make the model more complete. Through the calculation in the previous steps, we could get the scores of the indicators of Primorsky Krai and Jilin. Here, we introduced a comparative analysis model to make the analysis results clearer. Similarly, in the third part, we have added an explanation of the comparison results. The relevant indicators of the four subsystems are involved and further analyze the relevant results.

Point 3: Thirdly, in all research it is expected that there is a defined discussion of the results found to contrast with the theoretical and give more solid statements. This problem stems from the insufficient literature review. Likewise, the document lacks the conclusions where those relevant aspects of the work are consigned as well as its implications, limitations and future lines of research.

Response 3: Thank you very much for the suggestions on the research results. After team discussion and in-depth analysis, the content has been revised and improved in the manuscript.

We agree with you very much, in all research it is expected that there is a defined discussion of the results found to contrast with the theoretical and give more solid statements. Therefore, we have added Part 5. The conclusion of this article includes the conclusion, its impact, limitations, and future research directions.

According to the research process, this study draws four conclusions. First, there is a huge potential for cross-border cooperation between Primorsky Krai and Jilin. The policies of the two regions are in line with each other, and the cooperative economic foundation is good. Under the influence of geopolitical factors, the current cooperation opportunity is ripe. The development direction of cross-border cooperation between Primorsky Krai and Jilin is in terms of port logistics infrastructure cooperation, greenhouse agriculture and marine fishery trade cooperation, mineral resources development and deep processing cooperation, talent training, and cross-border labor cooperation. The first two conclusions are the direct results of the model analysis. Third, the development of cross-border regional cooperation is a key research issue under the trend of deepening the global division of labor and cooperation. The existing analysis of a single border region cannot fully support regional development. Therefore, it is necessary to combine the collaborative analysis of the corresponding region to scientifically assess the overall development potential of the region and provide scientific support for regional cooperation and development. This conclusion illustrates the innovation of research methods in this study. Fourth, this study could provide scientific support for promoting the two countries to strengthen cross-border cooperation and efficiently promote the construction and rapid development of economic corridors. The research on the cross-border development cooperation between Primorsky Krai and Jilin has put forward more targeted opinions and suggestions for the cross-border cooperation between China and Russia.

Finally, we summarized and discussed the existing problems and future research directions. The general laws of cross-border cooperation need to be sorted out and solved. For example, in the future, we will explore more regional cross-border economic cooperation laws and sort out universal theoretical laws; Whether there is a more advanced mode of cross-border cooperation, which makes the cooperation relationship truly stable, safe, and win-win. This study explores the economic development and transformation potential of the cross-border cooperation between Primorsky Krai and Jilin. It can provide basic scientific data support for the regional response to global changes and the research of China-Russia cross-border international cooperation, provide decision support for the revitalization strategy of Northeast China and the strategic cooperation of the development of the Far East of Russia, and make contributions to promoting the win-win cooperation between China and Russia. The evaluation of the development potential of cross-border cooperation hotspots has the value of the times and practical significance. The conclusion has strong scientific support for practical development.

Reviewer 3 Report

RE:Potential for Economic Transition and Key Directions of Cross-Border Cooperation between Primorsky Krai(Russia) and Jilin(China)

This paper is interesting. However, the authors may need to deal with the following issues before publication. 1. Show a descriptive table for variables in Table 1.

2. Hope to see quantitative analyses for the factors that drives the cross-border cooperation.

2. The current title for Table 1 is not proper.

3. The references on cross-border cooperation are not sufficient and reprehensive.  

4. Show a conclusion.

Author Response

We are very grateful for your comments regarding our manuscript. All your suggestions are very important to us, both for composing the manuscript and our further research. We have studied comments carefully and have made corrections which we hope meet with approval.

Point 1: Show a descriptive table for variables in Table 1. The current title for Table 1 is not proper.

Response 1: Thanks to peer reviewers for their advice on variables. There are many variables in this study. So we use text to explain the variables in the form. The relevant content has been added to the original manuscript. There are differences between China and Russia in statistical methods and caliber. This study will try to match the two similar indicators. the specific corresponding situation has been listed in Table 1. The relevant indicators of Primorsky Krai are on the left. And the relevant indicators of Jilin Province are on the right. The indicators of the target layer are composed of four components. The four indicators include economy, society, transportation and resources and environment. These indicators can summarize the overall socio-economic situation. They are also an important component of measuring the potential of economic transformation. The indicators of economic development level are represented by five indicators represented by the economic aggregate, It includes economic aggregates, economic quality, investment and trade, income and consumption and finance. And the level of social development reflects social services and social development through the six sectors represented by population. It includes population, urbanization, social burden, education, and medical care. The level of transportation reflects the strength and convenience of regional traffic. It includes the density of road network, traffic volume, and turnover. Resources and environment reflect the ecological advantages of resources in the region. To facilitate the calculation, the value data is used in this study. So we can ensure the consistency of domestic data. We will revise it based on following your suggestions to make the whole article easier to understand and clearer

We have revised the title of Table 1. And we are sorry for the error of not changing the template content. Thanks to the reviewers for their criticism and correction. We express our deep respect for your meticulous and serious work.

Point 2: Hope to see quantitative analyses for the factors that drives the cross-border cooperation.

Response 2: We are very grateful for your comments regarding our manuscript. We believe that quantitative analysis is an important part of cross-border cooperation research. This study quantifies the development potential of the coastal border area and Jilin Province. Thus, we can explore the factors that affect the economic development of the two cross-border cooperation zones. Our manuscript did not explain in detail the potential analysis results of promoting cross-border cooperation between Jilin and the coastal border area. We modified it.

We have added a comparative analysis of the development potential of cross-border regional cooperation between Primorsky Krai and Jilin. We have improved the development potential model of cross-border regional cooperation. In the third part, we modified the analysis content of relevant indicators of the four modules. In this study, we construct the evaluation index system of regional development potential from the perspective of cross-border cooperation. First of all, we collated the Primorsky Krai and Jilin field survey data, and Primorsky Krai with the government, the government of Jilin province, and other work.  Based on the content of the data, the geographical characteristics of the two regions, and the research direction of cross-boundary cooperation, it has established four subsystems including economic, social, transportation, resources, and environment, which are closely related to cross-border economic cooperation. At the same time, we consulted some experts in the process of determining indicators, trying to be scientific Finally, we established a multi-layer cross-border Regional development potential assessment (RDPA) model, which is composed of the target layer, factor layer, and index layer. In the next study, we will further explore cross-border economic and cooperation issues. In the future, we can determine the general model and coordination suggestions of the determinants of sustainable cross-border cooperation.

Point 3: The references on cross-border cooperation are not sufficient and reprehensive. 

Response 3: Thanks to the experts' suggestions to supplement the references in the cross-border cooperation area, we have revised this aspect in the manuscript.

The study on the development potential of cross-border regional cooperation is based on the study on the potential of regional development, with cross-border typical regions as its characteristics, and Primorsky Krai-Jilin as the key region for cross-border regional cooperation. We have added a literature review on the concept of cross-border co-operation, the importance of cross-border co-operation, the existing analysis of cross-border co-operation and its drivers, and as far as possible to take into account the adequacy of the reference and comprehensive.

In general, cross-border cooperation is one of the forms of territorial cooperation between different types of units in border areas. Common historical origins and other important similarities between adjacent areas on the border (for example, linguistic, cultural, constitutional, social, or economic) may contribute to uniting to achieve common development goals (Kurowska-Pys, 2016). Cross-border cooperation is a recognized approach to economic development in border areas (Lepik, 2009; Setnikar, 2013), based on the principle of equality of resources, responsibilities, risks, and interests (Aulakh, 1996). Brunet-Jailly (2022) reviews cross-border coordination and cooperation across continents and argues that regional drivers determine the type of relationship. Kurowska-Pysz (2017) carried out a comparative analysis of the study on the Polish-Lithuanian border and assessed factors for sustainable cross-border cooperation on the Czech-Polish border. Jose (2022) analyzes the cross-border governance of the Amazon cross-border region between Peru, Brazil, and Bolivia from a systematic perspective and explores the model of cross-border cooperation between Southern Finland and the Estonian border region. In general, cross-border cooperative research areas are concentrated in European countries. Cross-border differences between regions and the motivation for cross-border cooperation are to influence cross-border cooperation between regions (Castanho, 2017; Church, 1996; Löfgren,2008). There are strong differences in economic development between China and Russia, this difference has obvious complementary characteristics and goodwill to cooperate in cross-border economic cooperation, which determines the high research value of cross-border economic cooperation between the Primorsky Krai and Jilin province. At the same time, it is proved that the difference in the research area is the starting point of cross-border cooperative research, which makes full preparation for the next research. It is believed that the references in the field of cross-border cooperation have been supplemented in a more complete and comprehensive manner after revision.

Point 4: Show a conclusion.

Response 4: We would like to thank you for your comments on the findings of the study. We have also taken your valuable advice and added the chapter conclusion at the end of the article. The addition of this chapter also can not be separated from your previous suggestions, under which our team has a deeper understanding of this study. Finally, we conclude with four conclusions about this study.

First, there is huge potential for cross-border cooperation between Primorsky Krai and Jilin. The two regions have a well-matched policy and a Base of cooperation, and the time is ripe for cooperation under the influence of geopolitics. This conclusion runs through the full text of the narrative and can be said to be the main line of the study. The analysis result of the model has also been well demonstrated. Second, the Primorsky Krai of cross-border co-operation in Jilin is infrastructure co-operation, represented by the Port Logistics Industry, trade co-operation in Greenhouse Agriculture and Marine Fisheries, cooperation in Mineral Resources Development and intensive processing, talent development arising from population issues, and cross-border labor co-operation. These directions can provide some scientific suggestions for future cooperation between the two regions. Third, the development of cross-border regional cooperation is a key research issue under the trend of Deepening Global Division of Labor and cooperation. The existing analysis of a single border region does not adequately support regional development, and therefore needs to be combined with a synergistic analysis of the corresponding region of another country that is close to it to scientifically and fully assess the overall development potential of the region, to provide scientific support for the development of regional cooperation. For example, the analysis of cross-border cooperation between Primorsky Krai will not be persuasive if it is based solely on an independent analysis of the region. Who the Primorsky Krai should work with, how, and so on are all open questions. At this point, the cross-border Primorsky Krai area of Jilin was proposed separately, and the two regions were fully taken into account when establishing the model. Such cross-border synergy analysis results would be more scientific, targeted, and of more practical value. Fourthly, this study can provide scientific support for the two countries to choose key areas to strengthen cross-border cooperation and effectively promote the construction and rapid development of economic corridors. The Primorsky Krai is adjacent to Jilin province. These basic conditions have established the important status of the cross-border cooperation between the Russian Primorsky Krai and Jilin province. In the future, China and Russia will have more in-depth exchanges and cooperation here. This study provides more specific opinions and suggestions for sino-russian cross-border cooperation, and provides scientific and technological support for the promotion of sino-russian cross-border cooperation, the development of the sino-mongolia-russia economic corridor, and the implementation of the "Belt and Road" initiative.

Reviewer 4 Report

Dear authors,

This research paper describes topic – Potential for Economic Transition and Key Directions of Cross-Border Cooperation between Primorsky Krai (Russia) and Jilin (China). In their article authors notice, Regional economic transformation and development potential is an important evaluation criterion for measuring the existing and future development capacity of a region. As well, authors notice, that this study constructs a regional development potential assessment index system from the perspective of cross-border cooperation, establishes a multi-layer regional development potential assessment (RDPA) model of target layer-factor layer-indicator layer, and builds a development potential value system by coupling four modules of economic, social, transportation and resource environment.

And I would like to share with authors some doubts and remarks too: it seems important to notice, that it would be needed to concentrate on the discussion and conclusions of the study. It should be noted, that Conclussions section is missing here. Thus, when developing seperate sections of "Discussion" and "Conclusions" it would be needed to include to the debate more future oriented theoretical implications, thus accessing deeper discussion and concluding insights.

Author Response

We are very grateful for your comments regarding our manuscript. All your suggestions are very important to us, both for composing the manuscript and our further research. We have studied comments carefully and have made corrections which we hope meet with approval.

Point 1: It seems important to notice, that it would be needed to concentrate on the discussion and conclusions of the study. It should be noted, that Conclussions section is missing here. Thus, when developing seperate sections of "Discussion" and "Conclusions" it would be needed to include to the debate more future oriented theoretical implications, thus accessing deeper discussion and concluding insights.

Response 1: I would like to thank the peer reviewers for their professional comments on the findings of the study. After team discussion and in-depth analysis, the content has been refined in the manuscript. We add a more in-depth comparative analysis in the data analysis section and add a Conclusions section at the end of the article, summing up four related Conclusions. At the same time, we look forward to the future significance of the study, more complete and in-depth.

Four conclusions have been included in the manuscript. The first and second conclusions first two conclusions are closely related to the research work of this paper and are the direct results of model analysis. The Primorsky Krai of cross-border cooperation between Jilin and Hong Kong has huge potential. The two regions have a well-matched policy and a Base of cooperation, and the time is ripe for cooperation under the influence of geopolitics. The Primorsky Krai of cross-border co-operation in Jilin is infrastructure co-operation, represented by the Port Logistics Industry, trade cooperation in Greenhouse Agriculture and Marine Fisheries, co-operation in mineral resources development and intensive processing, talent development arising from population issues, and cross-border labor co-operation. The third conclusion is about the significance of this study in theory and method. The development of cross-border regional cooperation is a key research issue under the trend of deepening the global division of Labor and cooperation. The existing analysis of a single border region does not adequately support regional development and therefore needs to be combined with a synergistic analysis of the corresponding region with another country that is in cross-border proximity to it, to scientifically and fully assess the potential of regional development as a whole, and to provide scientific support for regional cooperation. In other words, this study, through the Primorsky Krai of cross-border research in Jilin province, proves that cross-border research should not only take into account the situation of the cross-border region itself but also focus on the adjacent regions of other countries, only by coordinating the analysis of the actual situation in the two countries and regions can we achieve a more comprehensive and accurate analysis. The last conclusion is about the practical significance of this study. In the context of globalization, exchanges and cooperation among countries have become increasingly frequent, which has created new demands for the evaluation of cross-border regional development potential. Our research can solve the problems related to cross-border regional cooperation from a more scientific perspective, which will also have a far-reaching impact on cross-border cooperation between the two countries and regional economic construction. This study can provide scientific support for the two countries to choose key areas to strengthen cross-border cooperation and effectively promote the construction and rapid development of economic corridors.

Round 2

Reviewer 1 Report

The article has been significantly improved in the literature review and research conclusions sections.

The study proposes priority directions for cross-border cooperation through the analysis of development potential. However, there seems to be a mismatch between the interpretation of the priority directions for cooperation and the results of the analysis of the selected development potential indicators. It is recommended that the authors summarize and emphasize the complementarity of regional indicators (regional differences) when proposing the focus directions and then elaborate on them in the context of regional realities.

Also, some of the language used in the article needs to be more international, e.g., "relevant domestic and foreign studies".

Author Response

Response to Reviewer 1 Comments

Dear Reviewer,

Thank you very much for your guidance and concern about our manuscript. From your guidance on many details, we are affected by your rigorous research attitude and scientific spirit. Meanwhile, thank you for your comments on our manuscript, they are all very valuable and very helpful for revising and improving our paper. It is our fortunate and honor to obtain high-level guidance from you. We have studied your comments carefully and have made correction point by point. The details are listed as follows. We sincerely hope that the revised manuscript can meet with your approval and be published in Sustainability. Thank you very much!

Sincerely,

Mengchen Ji

Point 1: The study proposes priority directions for cross-border cooperation through the analysis of development potential. However, there seems to be a mismatch between the interpretation of the priority directions for cooperation and the results of the analysis of the selected development potential indicators. It is recommended that the authors summarize and emphasize the complementarity of regional indicators (regional differences) when proposing the focus directions and then elaborate on them in the context of regional realities.

 Response 1:

Thank you very much to the reviewer for the comment on the fourth part of the article, " Cross-Border Cooperation Focus Direction"After conducting thorough research and conducting in-depth discussions as a team, we have made the necessary revisions.

In Section 4.1, we have identified the direction of cooperation between the port logistics industry and cross-border transportation infrastructure. Our analysis, based on model results, highlights that the ports of Primorsky Krai are advantageous projects. The development of high-quality ports in Primorsky Krai, specifically for cross-border cooperation, will create a strong demand for infrastructure. Meanwhile, our analysis of Jilin indicates that the region has a considerable disadvantage in terms of port shipping. However, China has extensive experience in the development of overseas infrastructure, providing a complementary opportunity for cross-border cooperation between the two regions.

In Section 4.2, we present a direction for cooperation between the agriculture and aquatic products industries. Our analysis is based on results derived from model calculations, which indicate that Primorsky Krai is endowed with abundant agricultural resources but lacks efficient exploitation methods such as agricultural mechanization. On the other hand, Jilin Province in China boasts a relatively mature experience in agricultural mechanization and is facing an increasing demand for agricultural imports. In terms of fisheries, Primorsky Krai is rich in fishery resources but faces difficulties in the development of its fishery industry due to its small scale and high transportation costs. Given its close proximity and high demand for fishery products, Jilin Province presents a promising opportunity for cooperation with Primorsky Krai in the trade of fishery products, leading to a reduction in the impact of transportation costs on prices and a mutually beneficial outcome. Furthermore, such cooperation will enable Primorsky Krai to better tap into the Chinese market and drive economic growth.

In Section 4.3, we present our views on the prospects of cooperation in mineral resource development and deep processing. The Far Eastern region of Primorsky Krai is rich in energy and minerals, with a mineral output ranking 38th in the country according to our model calculations. Despite its abundant resources, Primorsky Krai faces challenges in effectively developing its minerals, providing a strong potential for growth in the field. Meanwhile, the mines in Jilin Province are in the middle and late stages with a lack of resource succession, and there is a significant demand for major minerals. With Jilin Province's advantage in mining technology, both regions can complement each other and have a promising future in the field of mineral resource development and processing.

In Section 4.4, we see potential for talent training and cross-border labor cooperation between Primorsky Krai and Jilin Province. Our model analysis shows that both regions are facing population problems, with Primorsky Krai having a small labor force and Jilin Province facing an outflow of its population. Cross-border labor recruitment may be a solution for Primorsky Krai to address its labor shortage, while Jilin Province can recall its lost population and provide employment opportunities. The labor cooperation between the two regions would alleviate the labor shortage in Primorsky Krai and drive economic growth in both the Far East and Jilin Province.

Point 2: Also, some of the language used in the article needs to be more international, e.g., "relevant domestic and foreign studies".

Response 2: We appreciate the feedback provided by the reviewer. Following a comprehensive examination of high-quality literature, we have revised the wording of "relevant domestic and foreign studies" to the more formal and standardized term "Previous studies." Additionally, we have made general language revisions and improvements to the entire article to ensure conformity with academic standards. Thank you for your assistance in enhancing the quality of our work.

Reviewer 2 Report

A review of the titled work again shows that the authors have complemented several of the suggestions. However, there are several elements that need to be improved. In this sense, the document should be subjected to a major revision.

The inconsistency of the method of analysis persists. For example, it is mentioned that experts are consulted. How is the consultation method established? In the literature there are methods such as Delphi, TOPSIS and Expertones that allow the evaluation of the experts' answers.

Regarding the statistical method mentioned, there is none related to the known methods for its identification at a statistical level. The authors did not make the requested review on this aspect. Regardless of the formulas and methods used, credits should be given and the proponent should be asked who proposed them. If it is already a methodological proposal, this has an academic formalism and the respective credits are given. Therefore, the authors should make an effort to justify the method from the literature and not only mention that they build a model due to the complexity of integration of the variables. In the literature these methods are called aggregation operators or aggregation functions, I share with you two documents where you can find several methods related to the method used. Finally, it is imperative to correct this.

Blanco-Mesa, Fabio, Ernesto León-Castro, and José M. Merigó. 2019. "A Bibliometric Analysis of Aggregation Operators." Applied Soft Computing 81 (August). Elsevier: 105488. doi:10.1016/J.ASOC.2019.105488.

Calvo, Tomasa, Gaspar Mayor, and Radko. Mesiar. 2002. Aggregation Operators : New Trends and Applications. Physica-Verlag. https://dl.acm.org/citation.cfm?id=774556.

All equations are extensions of the text so they must have a comma or period at the end.

In the formulation of the equations there are inconsistencies. For example, in equations 4 and 5 there are indexes and subscripts that are not explained. This can be improved if they make the previously requested revision.

It was requested to create a section called discussion and it is not included in the text. They are reminded of the recommendation given: in all research it is expected that there is a defined discussion of the results found to contrast with the theoretical and give more solid statements.

The English should be revised in its entirety. There are syntax errors and it is weak in its structure and academic language.

Author Response

Response to Reviewer 2 Comments

Dear Reviewer,

Thank you very much for your guidance and concern about our manuscript. From your inquiry and guidance on many details, we are affected by your rigorous research attitude and scientific spirit. Meanwhile, thank you for your comments on our manuscript, they are all very valuable and very helpful for revising and improving our paper. Thank you very much for giving us the opportunity to revise the manuscript. It is our fortunate and honor to obtain high-level guidance from you. We have studied your comments carefully and have made correction point by point. The details are listed as follows. We sincerely hope that the revised manuscript can meet with your approval and be published in Sustainability. Thank you very much!

Sincerely,

Mengchen Ji

Point 1: The inconsistency of the method of analysis persists. For example, it is mentioned that experts are consulted. How is the consultation method established? In the literature there are methods such as Delphi, TOPSIS and Expertones that allow the evaluation of the experts' answers.

Response 1: Thank you for your comments.It is very important for revising and improving our manuscript. In the original manuscript, we did not explain the data collection and model development processes in detail. This was a mistake on our part.We sincerely apologize for this. Thank you for bringing up this point. We have made extensive changes to the revised manuscript, particularly to subsection 2 of section 2.2.2.

In the course of our research we sent research emails to a team of more than 10 Chinese and Russian collaborative research experts, including Dong  Suocheng, Director of the Center for Sustainable Development Research in Northeast Asia, Institute of Geographical Sciences and Resources, Chinese Academy of Sciences, and Professor Tsydypov Bair's team from the Far Eastern Branch of the Russian Academy of Sciences.

Our research team has conducted extensive fieldwork in Primorsky Krai and Jilin Province for previous studies.During our research, we also spoke with government experts and academics and gathered a wealth of first-hand practical information.

More than 10 experts were asked to rate the target layer weights' potential for growth online in the same comments set.The Delphi method was used in this study to figure out the weights based on what the experts said. We have added the corresponding calculation formula in line 242-249. Thank you for your guidance and concern about our manuscript.

Point 2: Regarding the statistical method mentioned, there is none related to the known methods for its identification at a statistical level. The authors did not make the requested review on this aspect. Regardless of the formulas and methods used, credits should be given and the proponent should be asked who proposed them. If it is already a methodological proposal, this has an academic formalism and the respective credits are given. Therefore, the authors should make an effort to justify the method from the literature and not only mention that they build a model due to the complexity of integration of the variables. In the literature these methods are called aggregation operators or aggregation functions, I share with you two documents where you can find several methods related to the method used. Finally, it is imperative to correct this.

Blanco-Mesa, Fabio, Ernesto León-Castro, and José M. Merigó. 2019. "A Bibliometric Analysis of Aggregation Operators." Applied Soft Computing 81 (August). Elsevier: 105488. doi:10.1016/J.ASOC.2019.105488.

Calvo, Tomasa, Gaspar Mayor, and Radko. Mesiar. 2002. Aggregation Operators : New Trends and Applications. Physica-Verlag. https://dl.acm.org/citation.cfm?id=774556.

Response 2:

Thank you for your comments. Your comments are very important for improving our paper. Our team has discussed and made the following changes.

Our model is generally based on our research team's previous findings. Li et al. (2019) conducted an assessment of the investment climate in Russia and constructed an ESI-PRA model, which was published in Chinese Geographical Science; Liu (2021) conducted an assessment of the investment climate in Mongolia, which was published in Geographical Research. Based on previous research findings, our study goes a step further, and we've constructed this model using a combination of first-hand information from our fieldwork and expert advice. Incorporating your suggestions, the mathematical methods used in this study include the aggregation function and the entropy method, and combined with the method by Yager to estimate the weights. We have added references to prove the origin of the method. In addition, we have included pertinent research examples from other scholars.Again, many thanks for your suggestions.

In response to your first comment, we have modified model in line 201-249.

Li, Fujia.; Liu, Qian.; Dong, Suocheng.; Cheng, Hao.; Li, Yu.; Yang, Yang.; Tsydypov, Bair.; Bilgaev, Alexey.; Ayurzhanaev, Alexander.; Bu, Xiaoyan.; Xia, Bing. Investment Environment Assessment and Strategic Policy for Subjects of Federation in Russia. Chinese Geographical Science, 2019, 29, 887-904. doi:10.1007/s11769-019-1051-1.

Liu, Qian.; Li, Fujia.; Zhuang, Yan.; Cheng, Hao.; Qi, Xiaoming.; Yang, Yang.; Ji, Mengchen. Investment environment assessment and investment strategy for provincial administrative units in Mongolia. Geographical research, 2021, 40, 11, 3046-3062.

Point 3: All equations are extensions of the text so they must have a comma or period at the end.

Response 3: Thank you for your comments. We have combined your suggestions with an extensive literature review. We find that the error you describe does exist. We apologize for the error in the details. We have corrected the content in the text.

Thank you for your review.

Point 4: In the formulation of the equations there are inconsistencies. For example, in equations 4 and 5 there are indexes and subscripts that are not explained. This can be improved if they make the previously requested revision.

Response 4: Thank you for your comments. I'm very sorry for the inconsistencie. We have errors of interpretation in the original manuscript. Our revisions are as follows:

In our formulas,  denoteis the target layer (q) initial score,  is standardized score of target layer, q indicates the target level code in development potential evaluation, item q element layer in the investment environment evaluation,,1 ≤ q ≤ 4. The  is the maximum value of the . The  is the minimum value of the .  is the weight of target layer (q).

The corresponding changes have been made to the manuscript. These modifications will clarify the formulas of this thesis, and we appreciate your feedback.

Point 5: It was requested to create a section called discussion and it is not included in the text. They are reminded of the recommendation given: in all research it is expected that there is a defined discussion of the results found to contrast with the theoretical and give more solid statements.

Response 5: We are very grateful for the comments of the reviewer. We have added a discussion section to the manuscript. We discuss three aspects of this study.

This study offers novel insights for future research on the development potential of cross-border regions. In this study, the cross-border region consists of distinct territories on both sides of the international boundary that belong to different nations. This is not a separate area. This viewpoint is innovative in comparison to previous studies. Most researchers would have chosen to explore cross-border economic development difficulties in a single region of a country. We examine both sides of the cross-border zone as a whole.

In addition, we discovered that field research is more important for practical cross-border collaboration problems. The information we obtained through comprehensive scientific research is more relevant to the actual situation of the study area as opposed to mere statistical analysis of data.  Many field research projects done by the Chinese Ministry of Science and Technology form the basis of this report. We have obtained a great deal of first-hand information. This information and statistical data have really aided our research.

Finally, we highlight the existing issues and future directions of study. We feel that the issues surrounding the general laws of cross-border cooperation must be resolved. In the future, for instance, we will investigate the laws of cross-border economic cooperation in more regions and sort out the universally applicable theoretical laws; whether there are more advanced models of cross-border cooperation to make the cooperation relationship truly stable, secure, and win-win, etc.

Point 6: The English should be revised in its entirety. There are syntax errors and it is weak in its structure and academic language.

Response 6: Thank you very much for your affirmation and guidance on our research, which is our greatest encouragement. According to your advice, we check carefully and improve the English grammar usage, structure and academic language.in our revised manuscript.

Your comments are all very valuable and very helpful for revising and improving our paper. It is our fortunate and honor to obtain high-level guidance from you.

Reviewer 4 Report

Congratulations for your efforts to review the article. It's seems better now. But still more theoretical insights are needed.

Author Response

Response to Reviewer 4 Comments

Dear Reviewer

Thank you very much for your encouragement! Your comments are all valuable and very helpful in our paper. We not only improved our manuscript greatly, but also perfected our whole research deeply. It is our fortunate and honor to obtain the high-level guidance from you. We have made a detailed revision in our revised manuscript, and the response is as follow. We sincerely hope the revised manuscript can gain your acceptance and be published in sustainability. Thank you very much!

Sincerely,

Mengchen Ji

Point 1: Congratulations for your efforts to review the article. It's seems better now. But still more theoretical insights are needed.

 Response 1: Thank you very much for your comments. We have added a discussion section to the manuscript. This will add more theoretical insights. We discuss three aspects of this study.

Firstly, this study provides new ideas for the research related to the development potential of cross-border regions. The cross-border area in this study includes different regions on both sides of the border line that belong to two countries. It is not a separate region. This point is an innovation for the previous studies. Most researchers would have selected a single region of a country to study cross-border economic development issues. We look at both sides of the cross-border region as a whole together.

In addition, we found the importance of field research to be higher for realistic cross-border cooperation issues. The information we obtained through comprehensive scientific research is more relevant to the actual situation of the study area as opposed to mere statistical analysis of data. This study relies on several field studies conducted by the Chinese Ministry of Science and Technology project. We have gained a large amount of first-hand information. These data have helped our study very well.

Finally, we summarize the current problems and future research directions. We believe that the problems about the general laws of cross-border cooperation need to be sorted out and solved. For example, in the future, we explore the laws of cross-border economic cooperation in more regions and sort out the theoretical laws with universality; whether there are more advanced models of cross-border cooperation to make the cooperation relationship truly stable, safe and win-win, etc.

Round 3

Reviewer 1 Report

The article has been greatly improved. It is recommended that it be accepted after further text formatting edits.

Author Response

Response to Reviewer 1 Comments

Dear Reviewer,

Thank you for taking the time to review our article and providing valuable feedback. We greatly appreciate your guidance throughout the review process, and we are thrilled to hear that our article has been greatly improved.

We would also like to express our gratitude for recommending that our article be accepted after further text formatting edits. We are eager to make these final edits and see our work published in Sustainability.

Once again, we thank you for your efforts in reviewing our article and providing constructive criticism. Your feedback has helped us to strengthen our research and presentation, and we are grateful for your support. Wish you every success in your work!

Sincerely, 

Mengchen Ji

Reviewer 2 Report

After reviewing the document, it can be seen that there are still some things to be corrected and that persist in the work, such as: 

All equations are extensions of the text so they must have a comma or period at the end.

The English should be revised in its entirety. There are syntax errors and it is weak in its structure and academic language. I suggest that document are revised by profesional editing English 

Author Response

Response to Reviewer 2 Comments

Dear Reviewer,

We would like to express our gratitude for your time and effort in reviewing our manuscript. Your insightful comments and suggestions have been invaluable in improving the quality of our work.

Thank you very much for giving us the opportunity to revise the manuscript. It is our fortunate and honor to obtain high-level guidance from you. We have carefully addressed the issues you raised in your review. Specifically, we have corrected all punctuation errors and have undertaken a thorough revision of the English language in the manuscript. "We have thoroughly revised the English language. Afterward, To ensure the highest quality of language, we have invited a native English speaker to perform a final editing of our manuscript. We have preserved the track changes from the final revision in the manuscript for the sake of clarity. We believe that these changes have significantly improved the manuscript.

Thank you again for your support and guidance in the review process. We sincerely hope that the revised manuscript can meet with your approval and be published in Sustainability. Thank you very much!

Sincerely,

Mengchen Ji
